# Systematic Study of Pressure Fluctuation in the Riser of a Dual Inter-Connected Circulating Fluidized Bed: Using Single and Binary Particle Species

**Yusif A. Alghamdi** [1,2,*] , **Zhengbiao Peng** [3,*] , **Caimao Luo** [4] , **Zeyad Almutairi** [1,5] , **Behdad Moghtaderi** [3] **and Elham Doroodchi** [3,6,*]

1   Sustainable Energy Technologies Center, College of Engineering, King Saud University, Riyadh 11421, Saudi Arabia; zaalmutairi@ksu.edu.sa
2   Deanship of Scientific Research (DSR), King Saud University, Riyadh 11421, Saudi Arabia
3   Priority Research Centre for Frontier Energy Technologies & Utilisation, the University of Newcastle, Callaghan, New South Wales 2308, Australia; Behdad.Moghtaderi@newcastle.edu.au
4   Design Confidence Consulting Company, Surry Hills, New South Wales 2010, Australia; caimao@designconfidence.com
5   Mechanical Engineering Department, King Saud University, P.O. Box 800, Riyadh 11421, Saudi Arabia
6   Priority Research Centre for Advanced Particle Processing and Transport, the University of Newcastle, Callaghan, New South Wales 2308, Australia
*   Correspondence: yalghamdi1@ksu.edu.sa (Y.A.A.); zhengbiao.peng@newcastle.edu.au (Z.P.); elham.doroodchi@newcastle.edu.au (E.D.); Tel.: +966-505666977 (Y.A.A.); +61-2-4033-9204 (Z.P.); +61-2-4033-9066 (E.D.)

**Abstract:** This study systematically investigates the pressure fluctuation in the riser of a dual interconnected circulating fluidized bed (CFB) representing a 10 kW$_{th}$ cold-flow model (CFM) of a chemical-looping combustion (CLC) system. Specifically, a single-species system (SSS) and a binary-mixtures system (BMS) of particles with different sizes and densities were utilized. The pressure fluctuation was analyzed using the fast Fourier transform (FFT) method. The effect of introducing a second particle, changing the inventory, composition (i.e., 5, 10 to 20 wt.%), particle size ratio, and fluidization velocity were investigated. For typical SSS experiments, the results were similar to those scarcely reported in the literature, where the pressure fluctuation intensity was influenced by varying the initial operating conditions. The pressure fluctuations of BMS were investigated in detail and compared with those obtained from SSS experiments. BMS exhibited different behaviour; it had intense pressure fluctuation in the air reactor and in the riser when compared to SSS experiments. The standard deviation (*SD*) of the pressure fluctuation was found to be influenced by the fluidization regime and initial operating conditions, while the power spectrum density (*PSD*) values were more sensitive to the presence of the particles with the higher terminal velocity in the binary mixture.

**Keywords:** circulating fluidized bed; chemical looping combustion; cold flow model; pressure fluctuation; riser; fast fourier transform; power spectrum density

## 1. Introduction

The application of gas-solid fluidized beds in industry is highly valued due to their abilities of providing an excellent interaction between solid particles and the gas medium, which in turn enhances energy conversion. Gas-solid fluidized beds are widely applied and have two main fields of application: (i) chemical engineering (i.e., catalytic cracking, mixing/segregation of powders), (ii) energy conversion (i.e., steam and hot water production in boilers) [1,2]. The hydrodynamics of gas-solid fluidized

beds are complex, primarily determined by the combined effects of solids' behaviour and bubbles' characteristics in terms of development, movement, and burst. The application of chemical-looping combustion (CLC) lies between these two points, i.e., it is both part of the chemical engineering application and in energy conversion and steam production.

It has been well documented that utilizing a mixture of metal oxides significantly improves the oxygen storage capacity of oxygen carrier particles in CLC systems that consist of dual interconnected circulating fluidized bed (CFB) [3,4]. This mixture of oxygen carrier was used as a single particles carrier containing both ingredients. However, other systems utilize a binary-mixture system (BMS) as two separate species that differ in sizes and/or densities in bubbling fluidized beds [5–16], and in a single-column fluidized bed [17–21]. Utilizing a BMS that differs in size and/or density will raise a number of operational uncertainties associated with the mixing/segregation of particles and the hydrodynamics of these complex systems. Mixing and segregation of binary solids that differ in both size and density have been investigated previously, where an operating map was developed to avoid any type of segregation (i.e., local or components segregation) in CLC systems [22–24]. The hydrodynamics of BMS in a dual interconnected CFB are yet to be investigated extensively. Specifically, the effects of parameters including particle size and density, mixture composition, total solids inventory (TSI) on the pressure profile, solids holdup and solid circulation rate have not been adequately examined [2,25–30]. In particular, studies concerning the role of the riser in determining the solid circulation rate, solids holdup and the stability of these CFBs, are very limited.

In addition, to investigate the hydrodynamics of CFB, time-series analysis of pressure fluctuation (gauge or differential pressure) or other signals such as solids holdup can also be used to describe the flow regimes [25,26,29,30]. Bai et al. (1996) studied the pressure fluctuation in a single column fluidized bed to characterise different fluidization regimes. They reported the solids holdup ($\phi_s$) values of $0.35 < \phi_s < 0.6$ for the bubbling fluidization, $0.15 < \phi_s < 0.35$ for the turbulent fluidization, $0.05 < \phi_s < 0.15$ for the fast fluidization and $\phi_s < 0.05$ for the pneumatic transport [25]. The standard deviation (*SD*) of pressure fluctuation was also used to determine the minimum fluidization velocity of the binary mixture of solids [31]. Lue and Wu (2000) used the sum of the *SD* of pressure fluctuation of each species to determine the *SD* of the binary mixture by multiplying the fraction of each species in the mixture to its corresponding individual *SD* under the same operating conditions [31]. Pressure fluctuation analysis has also been used to determine the combustion region in a fluidized bed reactor by studying the *SD* of the pressure fluctuation, and it was found that the combustion region was related to the regions that have high *SD* values [27]. In addition, the dominant power spectral density (*PSD*) is also helpful information for investigating the hydrodynamics in a fluidized bed with the method of fast Fourier transform (FFT). The irregular behaviour over time is due to the linear summation of periodic or random fluctuations, which is assumed by spectral and statistical analyses of electrical signals [28]. Conversely, owing to the complexity of gas–solid interactions, the hydrodynamics of the fluidized bed feature high non-linearity, anisotropy and different time scales that point to the non-stationary nature of the bed dynamics [1,32,33]. Therefore, the gas–solid fluidized bed has been considered as a chaotic system [34]. The *PSD* analysis by means of FFT analysis has been generally applied to the time series of pressure signals in a fluidized bed [35,36].

Most of the previous hydrodynamics studies have been focused on the SSS systems [37–44] with a view to gaining insight into: (i) the bed operating regime, (ii) solid entrainment, (iii) gas leakage, (iv) particle residence time, and (v) pressure profile [43,45,46]. The total solids inventory has been identified to play a significant role in defining the specific solid circulation rate whereby an increase in the inventory leads to an increase in solid circulation rate [23,45,47]. Also, attempts were also made towards: (i) improving the gas–solid interaction over the total height of the fuel reactor; (ii) reducing total inventory requirements of the fuel reactor by improving the solids-gas interaction; and (iii) increasing the cross-sectional area of the fuel reactor to provide steadier global solids holdup [43–45]. Operating the fuel reactor in the turbulent or fast fluidization regimes were also proposed as an

effective means to enhancing the gas–solid contact, thus to potentially reduce the total solids inventory which becomes of great relevance for increased plant capacities [43].

However, the riser is one of the major components of all CFB systems (such CLC) which contributes to the global solid circulation rate. The objective of this study is to investigate systematically the hydrodynamics of SSS and BMS in a cold-flow model (CFM) of CLC systems, in particular in the riser and its related component i.e., the air reactor, through a detailed pressure fluctuation analysis. The effects of the system key parameters including fluidization superficial gas velocity, particle size and density, mixture composition, total solids inventory, on the pressure fluctuation characteristics are investigated systematically based on the *PSD* and *SD* analyses.

## 2. Materials and Methods

Experiments were performed on a 10-kW$_{th}$ CFM–CLC system located at the Priority Research Centre of Frontier Energy Technologies and Utilisation at the University of Newcastle, Australia (Figure 1), with system total solids inventory capacity is between 1 to 3 kg. This is a lab scale 10 kW$_{th}$ CFM–CLC unit which was developed and designed based on Glickman scaling laws [48–50]. The total solids inventory was varied between 1 to 2 kg, which represents the specific inventory norms of 100–200 kg/MW$_{th}$ [39,51]. Particles physical properties, operating conditions, and operating dimensionless numbers are listed in Tables 1 and 2, respectively. The apparatuses' component geometry in terms of air and fuel reactors, risers and loop seals, and the applied experimental procedure, methodology and analysis in this work were described in greater detail in previous related studies (i.e., [23,52–54]).

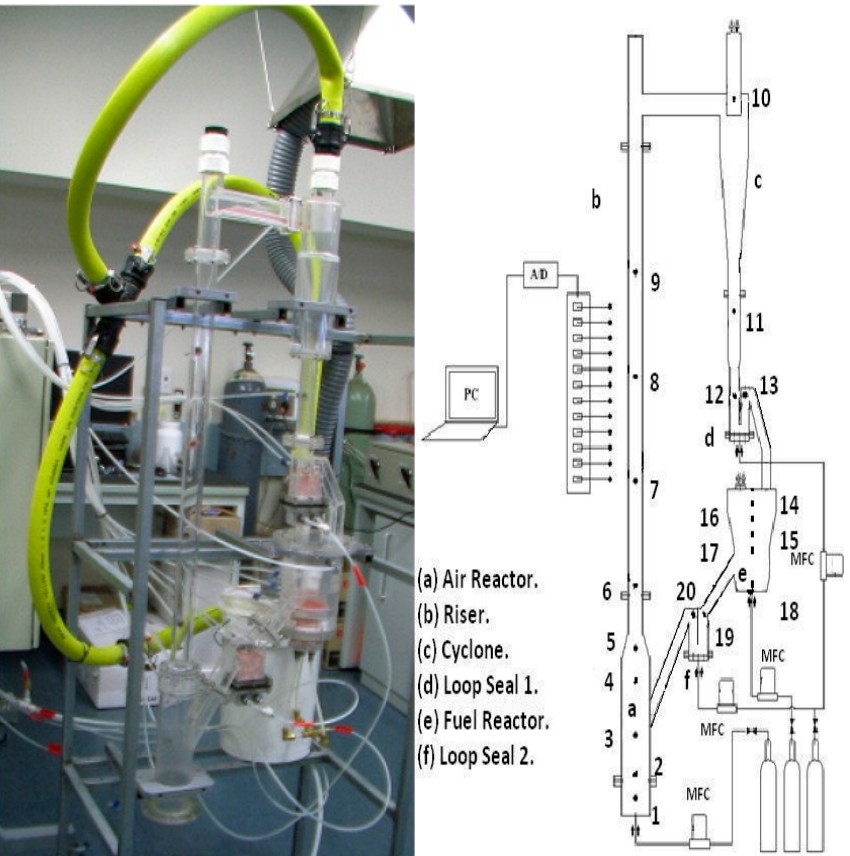

**Figure 1.** Schematic diagram of a 10-kW$_{th}$ cold-flow model–chemical-looping combustion (CFM–CLC) system. (The numbers shown here are the locations of the pressure ports) Reproduced with permission from [Alghamdi et al.], [Powder Technology]; published by [Elsevier], [2019] [48].

**Table 1.** Particles' physical properties.

| $d_p$, min (μm) | $d_p$, max (μm) | $d_p$, mean (μm) | $\rho_p$ (kg/m³) | $u_{mf}$ [a] (m/s) | $u_t$ [b] (m/s) | Particle Type |
|---|---|---|---|---|---|---|
| 90 | 106 | 98 | 2462 | 0.0085 | 0.58 | GB |
| 106 | 125 | 116 | 2462 | 0.012 | 0.75 | GB |
| 125 | 150 | 138 | 2462 | 0.016 | 0.99 | GB |
| 212 | 250 | 231 | 939 | 0.018 | 0.89 | PE |
| 250 | 300 | 275 | 939 | 0.025 | 1.1 | PE |
| 300 | 355 | 328 | 939 | 0.036 | 1.35 | PE |

GB: glass beads, PE: polyethylene, [a] calculated using Wen and Yu equation [55], [b] calculated using Haider and Levenspiel [56].

**Table 2.** Operating conditions for the single-species system (SSS) experimental cases for 10-kW$_{th}$ CFM–CLC.

| System Type | TSI (kg) | GB $d_p$ (μm) | $u_{g(A.R.)}$ (m/s) | $u_{g(A.R.)}/u_t$ | $u_{g(riser)}$ (m/s) | $u_{g(riser)}/u_t$ | $u_{g(F.R.)}$ (m/s) |
|---|---|---|---|---|---|---|---|
| SSS | 1 | 98 | 0.27 | 0.46 | 1.08 | 1.88 | 0.088 |
| SSS | 1 | 98 | 0.37 | 0.64 | 1.5 | 2.56 | 0.088 |
| SSS | 1 | 98 | 0.42 | 0.72 | 1.68 | 2.90 | 0.088 |
| SSS | 1.5 | 98 | 0.27 | 0.46 | 1.08 | 1.88 | 0.088 |
| SSS | 1.5 | 98 | 0.37 | 0.64 | 1.5 | 2.56 | 0.088 |
| SSS | 1.5 | 98 | 0.42 | 0.72 | 1.68 | 2.90 | 0.088 |
| SSS | 2 | 98 | 0.27 | 0.46 | 1.08 | 1.88 | 0.088 |
| SSS | 2 | 98 | 0.37 | 0.64 | 1.5 | 2.56 | 0.088 |
| SSS | 2 | 98 | 0.42 | 0.72 | 1.68 | 2.90 | 0.088 |
| SSS | 1.75 | 138 | 0.27 | 0.27 | 1.08 | 1.08 | 0.088 |
| SSS | 1.75 | 138 | 0.37 | 0.37 | 1.5 | 1.52 | 0.088 |
| SSS | 1.75 | 138 | 0.42 | 0.42 | 1.68 | 1.70 | 0.088 |
| SSS | 2 | 138 | 0.27 | 0.27 | 1.08 | 1.08 | 0.088 |
| SSS | 2 | 138 | 0.37 | 0.37 | 1.5 | 1.52 | 0.088 |
| SSS | 2 | 138 | 0.42 | 0.42 | 1.68 | 1.70 | 0.088 |
| SSS | 2.25 | 138 | 0.27 | 0.27 | 1.08 | 1.08 | 0.088 |
| SSS | 2.25 | 138 | 0.37 | 0.37 | 1.5 | 1.52 | 0.088 |
| SSS | 2.25 | 138 | 0.42 | 0.42 | 1.68 | 1.70 | 0.088 |

Glass bead (GB) particles with an apparent density of 2462 kg/m³ was used in the SSS experiments (Table 2); this is because it is the dominant species in the BMS experiments. GB particles Reynolds number in the air reactor ($Re_{dp(A.R.)}$) is between 1.75–3.8 and in the riser ($Re_{dp(riser)}$) is between 7–13.6. For the BMS, particles of different size GB and polyethylene (PE) with apparent densities of $\rho_{PE}$ = 939 kg/m³ as shown in Table 3 were used for the hydrodynamics studies (PE with $Re_{dp(A.R.)}$ = 4–6.42 and $Re_{dp(riser)}$ = 16.5–25.7). Different sizes of glass beads (GB) and polyethylene (PE) particles were prepared and used in the experiments, ranging from 98 to 328 μm, at three compositions of polyethylene particles, i.e., 5 wt.%, 10 wt.%, and 20 wt.%. A small amount of Larostat antistatic agent was added into the mixture to prevent electrostatic effects during fluidization. In the cold flow model, N$_2$ was used in the air reactor and air in the fuel reactor. The reason for using N$_2$ in the air reactor is to generate a stable fluidization of particles. This is because of the limited capacity of the available compressed air system (in term of supplied pressure), especially for the air reactor. Alternatively, N$_2$ was used that has very similar density and viscosity to those of air (i.e., $\rho_{(air)}/\rho_{(N_2)}$ = 1.04 and $\mu_{(air)}/\mu_{(N_2)}$ = 1.04) to supply the required pressure [48]. At room temperature in cold-flow model studies, it is believed that the error induced only by this subtle difference in density ratio would be negligible. The density ratios between the utilized particles and fluid ($\rho_p/\rho_f$) are 2113 and 806 for GB and PE, respectively.

**Table 3.** Operating conditions for the binary-mixtures system (BMS) experimental cases for 10-kW$_{th}$ CFM–CLC.

| System Type | Case No. | TSI (kg) | PE $d_p$ (μm) | GB $d_p$ (μm) | $d_{p(PE)}/d_{p(GB)}$ | PE Composition (wt%) | GB Composition (wt%) | $u_{t.H}/u_{t.L}$ | $u_{g(A.R.)}$ (m/s) | $u_{g(A.R.)}/u_{t.H}$ | $u_{g(riser)}$ (m/s) | $u_{(riser.)}/u_{t.H}$ | $u_{mf(mixture)}$ (m/s) [a] | $u_{g(F.R.)}$ (m/s) |
|---|---|---|---|---|---|---|---|---|---|---|---|---|---|---|
| BMS | 1 | 1 | 231 | 98 | 2.4 | 10 | 90 | 1.6 | 0.27 | 0.30 | 1.08 | 1.22 | 0.00769 | 0.088 |
| BMS | 2 | 1.5 | 231 | 98 | 2.4 | 10 | 90 | 1.6 | 0.27 | 0.30 | 1.08 | 1.22 | 0.00769 | 0.088 |
| BMS | 3 | 2 | 231 | 98 | 2.4 | 10 | 90 | 1.6 | 0.27 | 0.30 | 1.08 | 1.22 | 0.00769 | 0.088 |
| BMS | 4 | 1 | 275 | 98 | 2.8 | 10 | 90 | 2 | 0.27 | 0.25 | 1.08 | 0.98 | 0.00772 | 0.088 |
| BMS | 5 | 1.5 | 275 | 98 | 2.8 | 10 | 90 | 2 | 0.27 | 0.25 | 1.08 | 0.98 | 0.00772 | 0.088 |
| BMS | 6 | 2 | 275 | 98 | 2.8 | 10 | 90 | 2 | 0.27 | 0.25 | 1.08 | 0.98 | 0.00772 | 0.088 |
| BMS | 7 | 2 | 231 | 138 | 1.7 | 5 | 95 | 1.1 | 0.42 | 0.42 | 1.68 | 1.70 | 0.0150 | 0.088 |
| BMS | 8 | 2 | 231 | 138 | 1.7 | 10 | 90 | 1.1 | 0.42 | 0.42 | 1.68 | 1.70 | 0.0150 | 0.088 |
| BMS | 9 | 2 | 231 | 138 | 1.7 | 20 | 80 | 1.1 | 0.42 | 0.42 | 1.68 | 1.70 | 0.0150 | 0.088 |
| BMS | 10 | 2 | 231 | 116 | 2 | 10 | 90 | 1.2 | 0.27 | 0.30 | 1.08 | 1.20 | 0.0110 | 0.088 |
| BMS | 11 | 2 | 231 | 116 | 2 | 10 | 90 | 1.2 | 0.37 | 0.42 | 1.5 | 1.70 | 0.0110 | 0.088 |
| BMS | 12 | 2 | 231 | 116 | 2 | 10 | 90 | 1.2 | 0.42 | 0.47 | 1.68 | 1.89 | 0.0110 | 0.088 |
| BMS | 13 | 2 | 328 | 138 | 2.4 | 10 | 90 | 1.4 | 0.27 | 0.20 | 1.08 | 0.80 | 0.01512 | 0.088 |

[a] Calculated using Cheung et al. equation [57].

CFM–CLC are usually made out of Perspex material, therefore, there are two important reasons of not using active metal oxide in CFM–CLC units: (i) abrasion effect if optical measurement methods were utilised, and (ii) the expensive use of active metal oxides in the CFM–CLC experiments. In the hydrodynamic studies of fluidized bed (specially in CLC systems) dimensionless analysis is conducted, which is essential for selecting special particles, these particles should relatively represent the same physical features in terms of their sizes and densities as those of metal oxide particles used in an actual CLC process [43,52]. This dimensionless analysis considers the density and the viscosity of the fluid at the required temperature. The selected particles were calculated based on the Archimedes number (*Ar*) expressed as:

$$Ar = \frac{\rho_f\left(\rho_P - \rho_f\right)gd_P^3}{\mu^2} \tag{1}$$

where $d_P$ is the particle size, $g$ is the acceleration due to gravity, $\mu$ is the gas viscosity, $\rho_f$ and $\rho_P$ are the fluid (i.e., gas) and particle densities, respectively. By assuming that the dimensionless *Ar* number for the CFM at 25 °C is equal to the *Ar* number at the actual hot CLC process (i.e., 500–1000 °C):

$$Ar_{(25°C)} = Ar_{(hot)} \tag{2}$$

where $Ar_{(25\,°C)}$ was determined using $\mu$ and $\rho_f$ for air at 25 °C, $\rho_P$ and $d_P$ of GB and PE, and $Ar_{(hot)}$ is expressed as,

$$Ar_{(hot)} = \frac{\rho_{f(hot)}\left(\rho_P - \rho_{f(hot)}\right)gd_P^3}{\mu_{(hot)}^2} \tag{3}$$

By substituting Equation (3) into Equation (2),

$$Ar_{(25°C)} = \frac{\rho_{f(hot)}\left(\rho_P - \rho_{f(hot)}\right)gd_P^3}{\mu_{(hot)}^2} \tag{4}$$

The diameter of the metal oxide in the actual CLC process can be obtained by knowing the metal oxide density, which is well defined in the literature, and by knowing the density and the viscosity of the fluid at the required temperature. The diameter of the metal oxide in the actual CLC process, which represents the particles used in the CFM–CLC system in terms of density and size, can be found using the following expression:

$$d_P = \left[ \frac{Ar_{(25°C)}\mu_{(hot)}^2}{\rho_{f(hot)}\left(\rho_P - \rho_{f(hot)}\right)g} \right]^{\frac{1}{3}} \tag{5}$$

Therefore, based on this analysis, 98–138 µm (GB) at ambient operating conditions (i.e., 25 °C) relatively represents particles such as CuO ($\rho_P$ = 6315 kg/m$^3$, 117–313 µm), Fe$_2$O$_3$ ($\rho_P$ = 5242 kg/m$^3$, 125–330 µm), NiO ($\rho_P$ = 6670 kg/m$^3$, 115–307 µm) and SiO$_2$ ($\rho_P$ = 2684 kg/m$^3$, 156–417 µm) at 500–1000 °C using air in the air reactor (*Ar* = 78–217) and CH$_4$ in the fuel reactor (*Ar* = 48–133). Similarly, 234–275 µm (PE) at ambient operating conditions (i.e., 25 °C) relatively represents particles such as CuO ($\rho_P$ = 6315 kg/m$^3$, 203–457 µm), Fe$_2$O$_3$ ($\rho_P$ = 5242 kg/m$^3$, 216–487 µm), NiO ($\rho_P$ = 6670 kg/m$^3$, 200–449 µm) and SiO$_2$ ($\rho_P$ = 2684 kg/m$^3$, 270–608 µm) at 500–1000 °C using air in the air reactor (*Ar* = 403–1111) and CH$_4$ in the fuel reactor (*Ar* = 247–679). Under constant *Ar* number, to have hydrodynamic similarity, the Froude number ratio (i.e., between the hot and cold model) can be close to 1 [43,48–50]. The velocity of the air reactor in the hot model should be between 0.9–1.8 times that of the CFM. The applicability of the result obtained from this lab-scale CFM–CLC using the particles in Table 1 is validated and might only be applied to this or similar units. The scaling from 10 kW$_{th}$ CFM–CLC (i.e., lab scale) to 200 kW$_{th}$ CFM–CLC (i.e., demonstration pilot plant) using similar particles and approach was also validated in previous work and showed hydrodynamic similarity [48].

At fixed air reactor fluidization velocity ($u_{g(A.R.)}$), each case was conducted at least 3 times with five fuel reactor fluidization velocities ($u_{g(F.R.)}$) in the range of 0.0294–0.1470 m/s. After conducting the experiments and steady-state operating conditions being reached (Figure A1), the pressure readings were taken using 20 pressure ports allocated on the apparatus (i.e., Honeywell micro switch sensing and control, 142PC05D, 164PC01D37 and 142PC01D, Morris Plains, New Jersey, USA) that recorded pressure at different points in the system, and in the same time the solid circulation rate was measured. The solid circulation rate (i.e., g/min) was obtained using the direct measurement method [53] during a steady state operation by stopping the aeration of loop seal 1 and measuring the time required to reach a particle bed height accumulation of 1–2 cm (repeated for 3 times). The solids holdup was obtained using pressure drop measurement (i.e., $\phi_s = \Delta p/[\rho_p g \Delta h]$) from the pressure ports installed along the CFM–CLC system (the numbers in Figure 1 are the location of each pressure port). Two types of pressure data throughout the CFM–CLC system were recorded simultaneously at steady-state operation conditions (i) an averaged pressure value from 100 readings of each pressure port (i.e., 10 readings per 1 s for 10 s interval), and (ii) transient pressure data recorded for each port with a rate of 500 readings per second for a period of 10–60 s (i.e., 5000–30,000 readings). The pressure at each point in the system was compared with the pressure values of the zero reference point measurements that were taken at the beginning of each experiment. Thus, the absolute pressure values of the system could be obtained. The confidence intervals of the error associated with the pressure measurements were less than ±0.03 kPa at 95%, where also the uncertainty analysis in our study showed that the associated error of the solid circulation rate measurements was rather reasonable, falling within 5%–15% [52,53].

To understand the effect of using BMS in the CFM–CLC, a systematic investigation was conducted using firstly SSS, and later another species was added into the system for BMS analysis. For both systems, the effect of changing the superficial gas velocity i.e., in the air reactor ($u_{g(A.R.)}$) and riser ($u_{g(riser)}$), total solids inventory, particle size, along with the effect of adding another species at constant fuel reactor $u_{g(F.R.)}$ on riser pressure fluctuation were analysed qualitatively and quantitatively. In the CFM–CLC, the air reactor has larger diameter (i.e., $d_{in(A.R.)} = 80$ mm and $h = 300$ mm) and operates under turbulent to fast fluidization regimes (to increase the particle residence time in the reactor), while the riser has smaller diameter (i.e., $d_{in(riser)} = 40$ mm and $h = 1150$ mm) and operates under pneumatic conveying regime to give the particles that departed the air reactor the additional momentum to circulate the system, as shown in Figure 1. For the first part of the experimental investigation, the pressure fluctuation was taken for 60 s (i.e., for SSS). However, to save experimental time and since there is no change in the global solid circulation in the system at steady state, it was decided to be taken at the same sampling rate, however, only for 10–15 s. It is worth noting that the transient pressure fluctuation figures are averaged for every 60 readings to avoid a thick line that cannot be read and compared, all pressure fluctuation figures are produced for 11–12 s. The averaged pressure, solids holdup, and solid circulation rate profiles of the examined conditions along with their detailed discussions can be found in our earlier studies [23,52].

*Pressure Fluctuation Analysis*

The complex Fourier method was used in the analysis of the pressure signals monitored in the air reactor and the riser. For the complex Fourier method, the first step is to find the complex Fourier expansion amplitudes for given temporal-varying pressure:

$$\Delta P(t) = X_0 + \sum_{i=1}^{N} [X_i \cos(\omega_i t) + Y_i \sin(\omega_i t)] \tag{6}$$

$$PSD(\omega_i) = \frac{X_i^2 - Y_i^2}{2\pi} \tag{7}$$

in which $t$ denotes to the time of the monitored variable, $\omega_i$ is the angular frequency ($\omega_i = i \times (2\pi/P)$), $P$ is the monitored period ($P = 11.0$ s), $i$ is an integer and $X$ and $Y$ are the amplitude at the angular frequency $\omega_i$.

The nature of the Fourier analysis as shown in Equation (6) is to fit all monitored data using an analytic function expressed as the right hand side (RHS) of Equation (6). There are 2N + 1 unknowns ($X_0, X_1, X_2 \ldots X_N; Y_1, Y_2, \ldots Y_N$) which are solved by the Fourier series analysis. During 11 s period, 5500 monitored points match well with the prediction from the Fourier coefficients ($X_0, X_1, X_2 \ldots X_N; Y_1, Y_2, \ldots Y_N$) as shown by Figure 2.

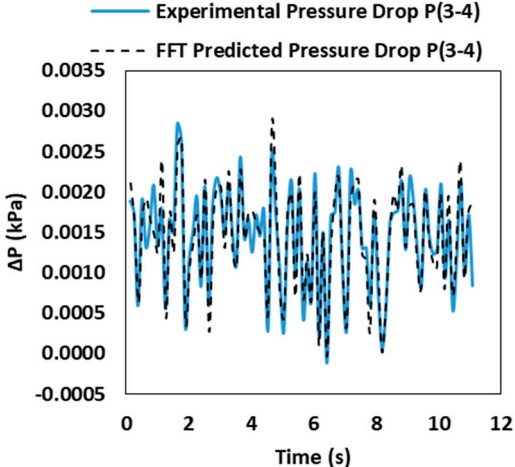

**Figure 2.** Comparison between the experimental pressure drop against fast Fourier transform (FFT) predicted values.

The signal-sampling rate for the pressure fluctuation was $500 \text{ s}^{-1}$ over a time period of 10–60 s at the steady-state operation. The pressure fluctuation *SD* was calculated by:

$$SD = \left[ \frac{1}{N} \sum_{i=1}^{N} \left( \Delta P_i - \Delta P_{Average} \right)^2 \right]^{\frac{1}{2}} \tag{8}$$

where $N$ is the number of the sampling points; $\Delta P_i$ is the pressure drop for each sampling point and $\Delta P_{Average}$ is the average pressure drop over the entire sampling points. Some of the *SD* data are found to be slightly higher or lower due to the artefact of the local solids holdup just before or after the area of interest (i.e., point 3–4 in the air reactor and point 6–9 in the riser). Thus, for some points the average of three *SD* values was taken (e.g., *SD* for pressure at point 6, *SD* for pressure at point 9 and *SD* for pressure drop point 6–9).

## 3. Results and Discussion

### 3.1. Single Species and Binary Mixtures Systems (BMS)

The effect of adding a second species to the total solids inventory was investigated, the study consisted of adding polyethylene particles with apparent density of 939 kg/m$^3$ to the binary mixture. The segregation intensity between/within the air and the fuel reactors (i.e., the local and components segregation) are studied in greater detail in [23,24]; in this work the discussion is more focused on the pressure fluctuation analysis of BMS.

Adding a second species to the inventory affected the overall pressure and solids holdup profiles, however, the profiles in general were found to be comparable to SSS [23,52]. Higher riser pressures and hence a greater solids holdup and solid circulation rate are obtained when using a BMS. The pressure fluctuation in the air reactor and the riser are also greater than that of the SSS, indicating some

discrepancies in the local solids holdup profile for the BMS. That is because using a second species in the system with a larger size ($d_p$), a lower density and a higher terminal velocity increases the overall bed height and the solids holdup of that species in the system. In response, the fluctuation intensity in terms of *SD* and its corresponding *PSD* increases as well. Therefore, increasing the holdup in the air reactor between points 3 and 4 increases the interaction and collision of particles. As shown in Figure 3a,b, the pressure fluctuation value and intensity, and the *PSD* are higher at the air reactor for the BMS than that of SSS.

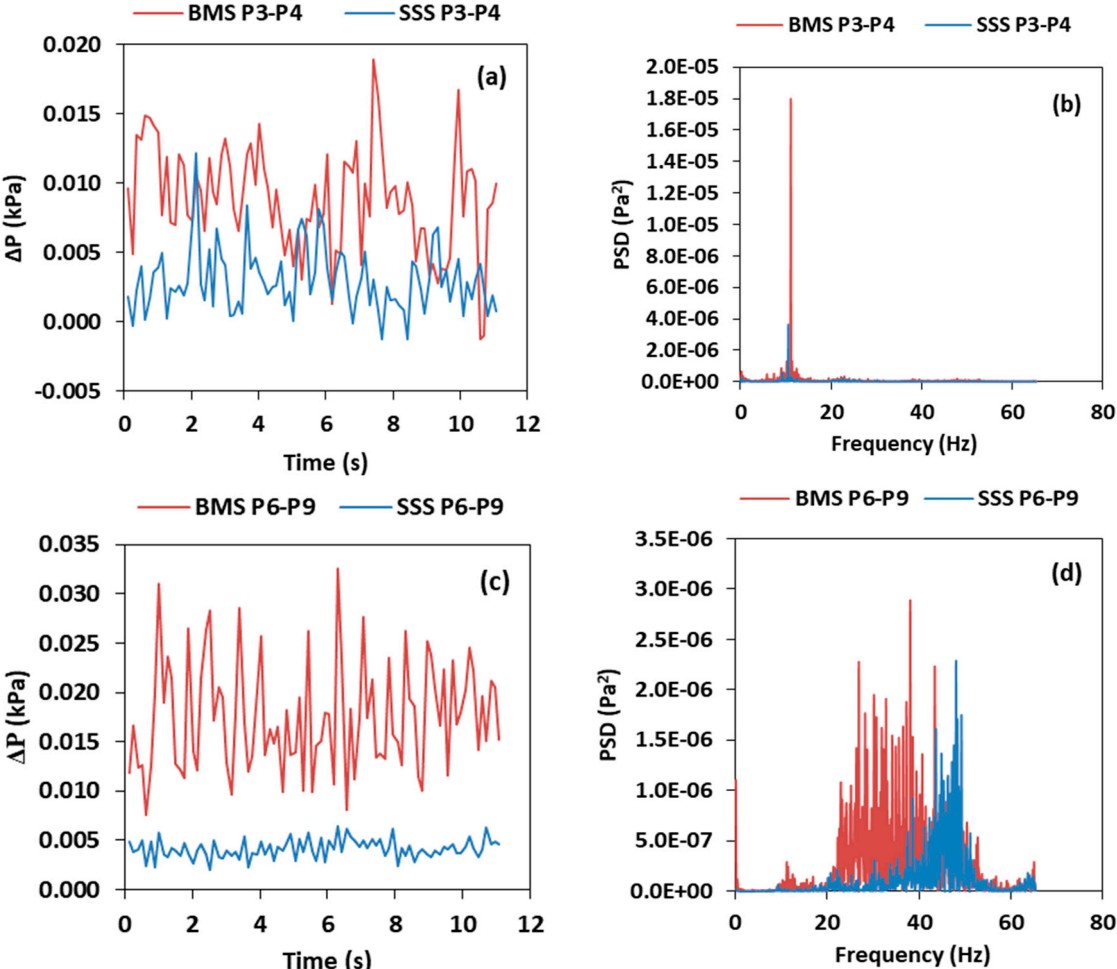

**Figure 3.** Evolution of the pressure fluctuation and the spectral analysis on the air reactor (**a**,**b**) and the riser (**c**,**d**) for SSS [$d_{GB}$ = 98 μm, $u_{g(riser)}$ = 1.08 m/s and $u_{g(F.R.)}$ = 0.088 m/s] and BMS [10 wt.% $d_{PE}$ = 275 μm, 90 wt.% $d_{GB}$ = 98 μm, $u_{g(riser)}$ = 1.08 m/s and $u_{g(F.R.)}$ = 0.088 m/s] at total solids inventory (TSI) = 1.5 kg.

In the riser, some of the particles are observed to fall back as they reach the top of the riser, resulting in a back mixing near the wall. Therefore, the riser also follows the same behaviour where the pressure fluctuation intensity and the *PSD* are higher for BMS, Figure 3c,d. The wide and strong periodicity of the pressure fluctuation in the riser (i.e., Figure 3c) which demonstrates itself as multiple narrow peaks (Figure 3d) of *PSD* indicating the presence of solids holdup bubbles of different sizes in the riser. This is the artefact of falling particles adjacent to the riser wall which causes a large variation in the local solids holdup when it is mixed with the upcoming flow from the air reactor. Decreasing the polyethylene size from 275 μm to 231 μm shows the same trend where the BMS has a higher pressure fluctuation intensity and *PSD* in the air reactor and the riser. This intensity (as discussed above) manifests itself as stronger *SD* and a higher dominant *PSD* amplitude (Figure 4). It is worth noting that at 1 kg, the values of *SD* and *PSD* were close, and this is because at this inventory the volume of the 10 wt.% of

PE particles was not high enough to show the difference, however, it was more pronounced at higher inventory. Nevertheless, it still shows a slight increasing trend. In the air reactor at 1 kg for SSS, BMS (case 1–3) and BMS (case 4–6) the values of *SD* were 13.24, 13.55 and 16.78 (Pa), and the values of *PSD* were 1.9, 1.53 and 5.7 ($\times 10^{-7}$ Pa$^2$), respectively. While in the riser, the values of *SD* were 9.2, 9 and 10.77 (Pa), and the values of *PSD* were 6.2, 5.40 and 6.7 ($\times 10^{-7}$ Pa$^2$), respectively.

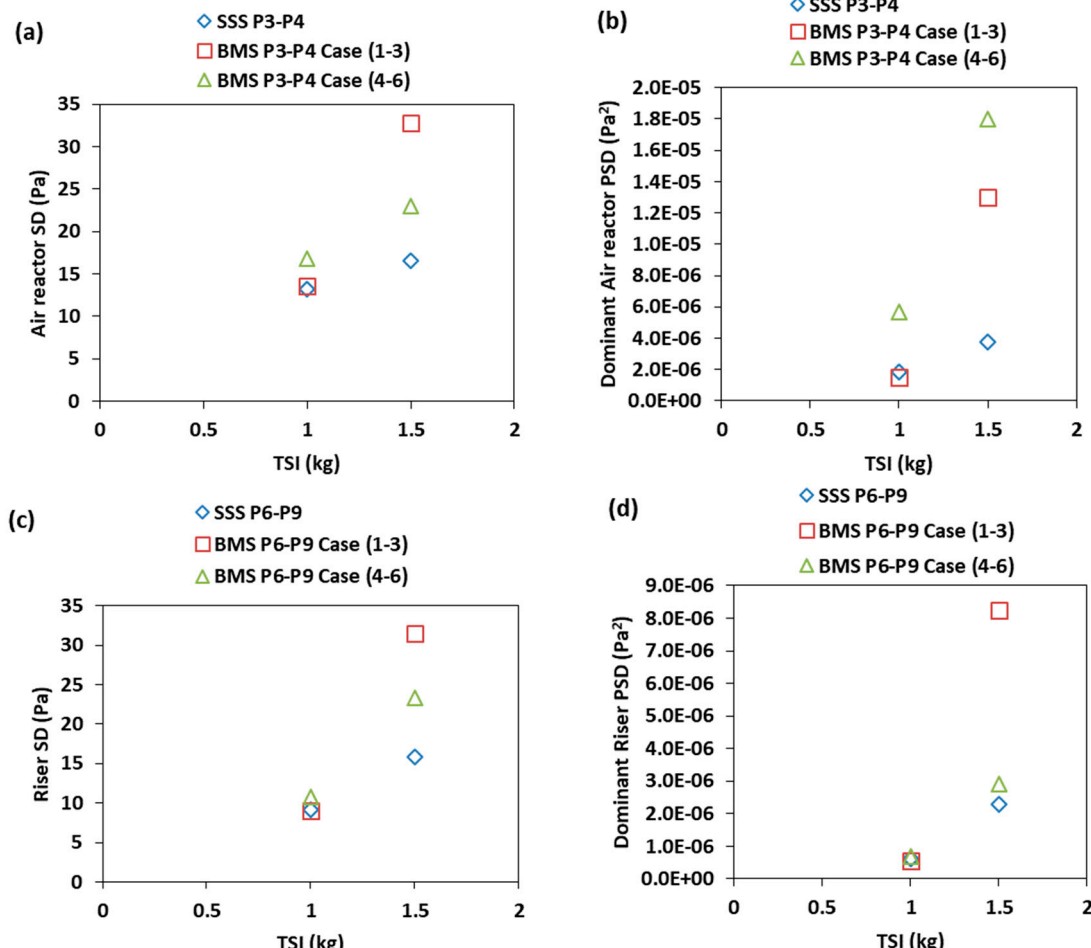

**Figure 4.** Comparison of the standard deviation (*SD*) of the pressure fluctuation and the power spectral density (*PSD*) analysis on the air reactor (**a**,**b**) and the riser (**c**,**d**) between SSS [$d_{GB}$ = 98 μm] and BMS [10 wt.% $d_{PE}$ = 231 μm (case 1–3) and 275 μm (case 4–6), 90 wt.% $d_{GB}$ = 98 μm] at $u_{g(A.R.)}$ = 0.27, $u_{g(riser)}$ = 1.08 m/s and $u_{g(F.R.)}$ = 0.088 m/s.

### 3.2. Effect of the Superficial Gas Velocity

The effect of the superficial gas velocity ($u_{g(A.R.)}$ and $u_{g(riser)}$) on the pressure fluctuation for SSS and BMS were studied along the air reactor and the riser. Changing $u_{g(A.R.)}$ at a constant total solids inventory influenced the overall pressure and solid circulation rate profiles [23,52], pressure fluctuation, the frequency distribution and the amplitude of that fluctuation in terms of *SD* and *PSD* between points 3 and 4 in the air reactor and between points 6 and 9 in the riser.

For SSS, at a lower $u_{g(A.R.)}$, the variation of the solids holdup is more significant in the air reactor than that at a higher velocity, as shown in Figure 5a,b. The BMS followed the same behaviour in the air reactor as those of the SSS when the inlet velocity increased from 0.27 to 0.42 m/s, Figure 5c,d. This is because at a low gas inlet velocity, the solids holdup for both systems i.e., SSS and BMS are higher. In general, the pressure fluctuation is less intense with smaller dominant amplitude in terms of *PSD* and frequency distribution at higher velocities, as shown in Figure 5b,d. This is because as the velocity increases in the air reactor, the solids interaction and holdup decrease. This behaviour was

also observed by Bai et al. (1996) when operating the CFB under the same operating regimes in the air reactor [25]. Therefore, the strong periodicity of pressure fluctuation that manifests itself as multiple narrow peaks of PDS indicating the presence of bubbles of different sizes in the air reactor. This is reasonable since the air reactor is operating under turbulent to fast fluidization regime. The value of this amplitude will become less as the air reactor fluidization velocity increased [28].

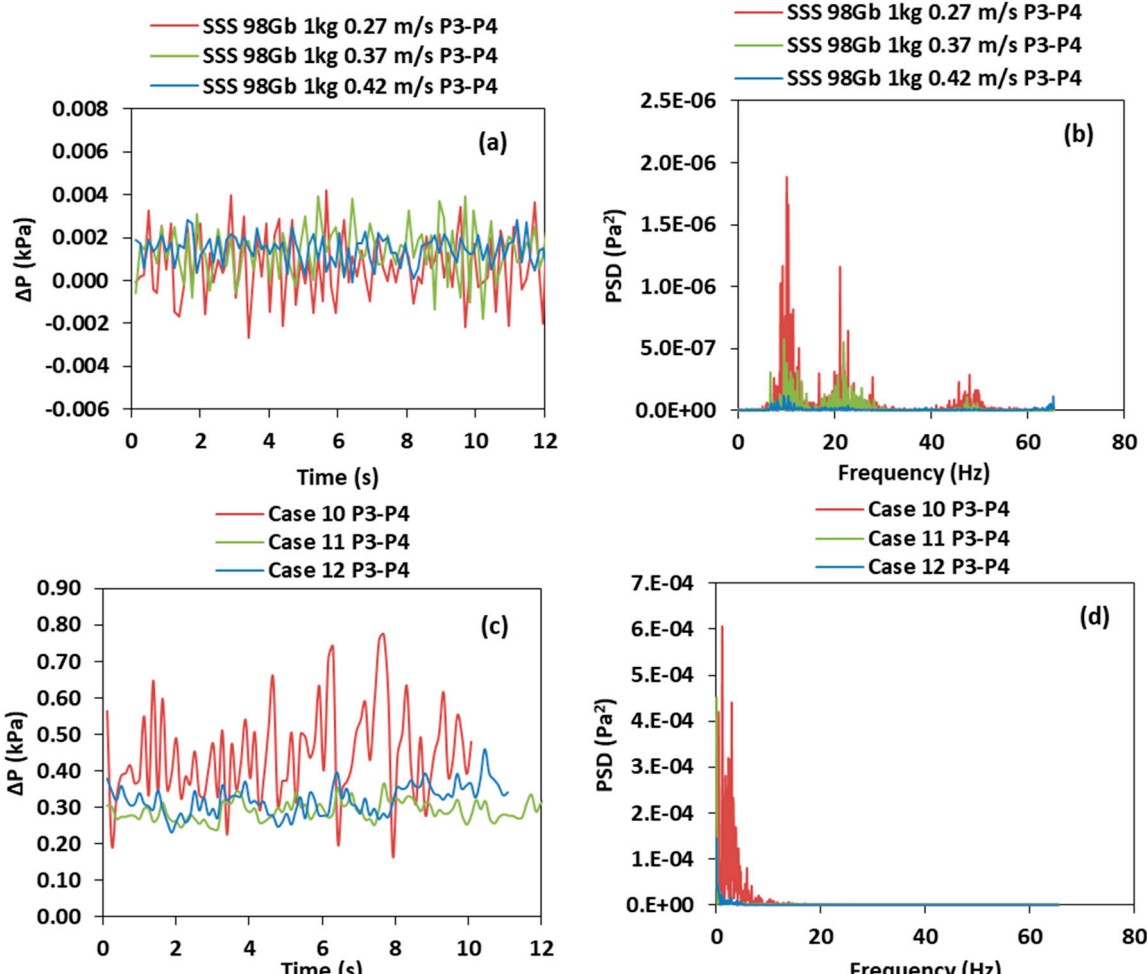

**Figure 5.** Evolution of the pressure fluctuation and the spectral analysis at the air reactor between points 3–4: (**a,b**) of SSS [$d_{GB}$ = 98 μm and TSI = 1 kg] and (**c,d**) of BMS [10 wt.% of $d_{PE}$ = 231 μm, 90 wt.% of $d_{GB}$ = 116 μm, TSI = 2 kg] as a function of $u_{g(A.R.)}$ = 0.27 m/s, 0.37 m/s and 0.42 m/s, and $u_{g(F.R.)}$ = 0.088 m/s.

In the riser between points 6 and 9, the solids holdup and the pressure values are higher at higher inlet velocity. This is because increasing the superficial gas velocity leads to the conveying of more solids to the riser, as shown in Figure 6a. For SSS, the dominant amplitude is higher at a lower velocity [28]. The reason behind this is that, at a lower velocity, more solids falls downwards near the riser wall [53]; consequently, more disturbances are observed in the riser due to this chaotic behaviour. Therefore, in the riser, always one narrow high peak is observed along with a few smaller ones; this is because the riser's main function is to transport the solid particles to the cyclone and then to the fuel reactor. Therefore, if the superficial gas velocity in the riser is not high enough to convey all particles out of the riser, the interactions between particles will become more intense, and will show consecutively as intense pressure fluctuations, presenting as multiple smaller peaks. The frequency distributions for 1.08 and 1.50 m/s in the riser are wider and more intense than that at 1.68 m/s, as shown in Figure 6b.

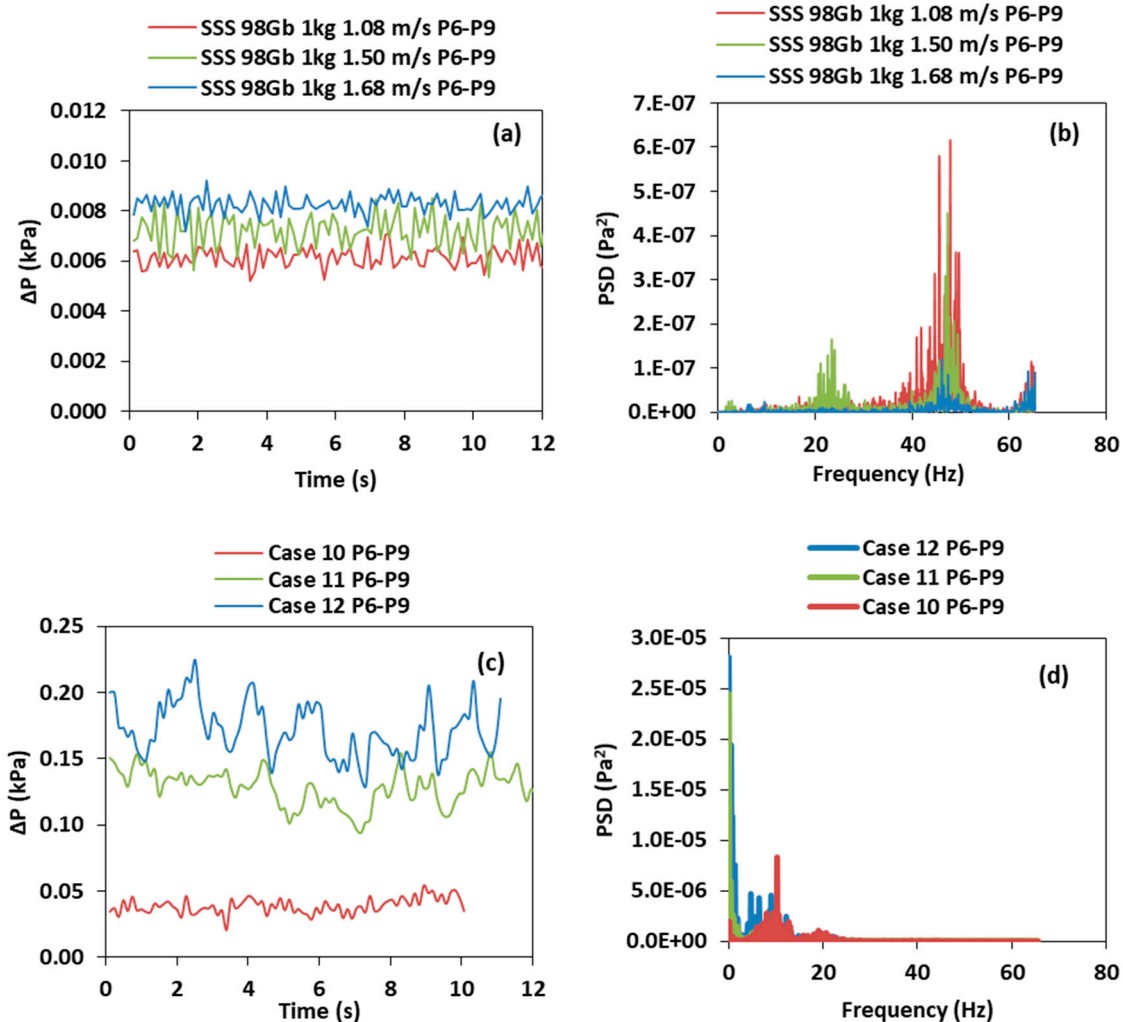

**Figure 6.** Evolution of the pressure fluctuation and the spectral analysis at the riser between points 6–9: (**a**,**b**) of SSS [$d_{GB}$ = 98 μm and TSI = 1 kg] and (**c**,**d**) of BMS [10 wt.% of $d_{PE}$ = 231 μm, 90 wt.% of $d_{GB}$ = 116 μm, TSI = 2 kg] as a function of $u_{g(riser)}$ = 1.08 m/s, 1.50 m/s and 1.68 m/s, and $u_{g(F.R.)}$ = 0.088 m/s.

Similarly, for BMS, the pressure fluctuation value in the riser is higher at a higher superficial gas velocity, see Figure 6c. This is because more particles are transported to the riser and, therefore, it causes the solids holdup to increase, especially for the larger size in the BMS (i.e., PE particles). As the riser velocity increased from 1.08, 1.5 to 1.86 m/s, its ratio to the terminal velocity of the particle with the higher terminal velocity in the BMS increased from 1.2, 1.7 to 1.89, respectively. Accordingly, the solid circulation rate increased rapidly from 90, 237.7 to 364.1 g/min. Even though the percentages that circulated in the loop for glass beads and polyethylene in the total solid circulation rate for these cases were constant and around 92.8% and 7.2%, respectively. The circulation of the polyethylene particle that has a higher terminal velocity in the BMS increased from 6.81, 15.77 to 27.16 g/min [23], respectively. However, as shown in Figure 6d, the *PSD* behaviour in the riser for the BMS was opposite to that observed in SSS, this is because of the increased presence of the larger size particles (i.e., has higher $u_t$), thus, the intensity and disturbance at higher velocity was observed.

In the air reactor for SSS and BMS, Figure 7a,c show the effect of varying $u_{g(A.R.)}$ at a constant total solids inventory on the *SD* and *PSD* of the pressure fluctuation. The *SD* and *PSD* values decreases as the velocity increases due to the same reasons mentioned in the above discussion. In the air reactor for BMS the *SD* and the *PSD* decreased as the inlet velocity increased and followed the same behaviour as that of SSS. Increasing the air reactor superficial gas velocity in the BMS increases the solids holdup of larger particles in the riser. The *SD* of the pressure fluctuation and its corresponding *PSD* of the

dominant amplitude increased and followed the opposite behaviour (Figure 7d) when compared with the SSS (Figure 7b).

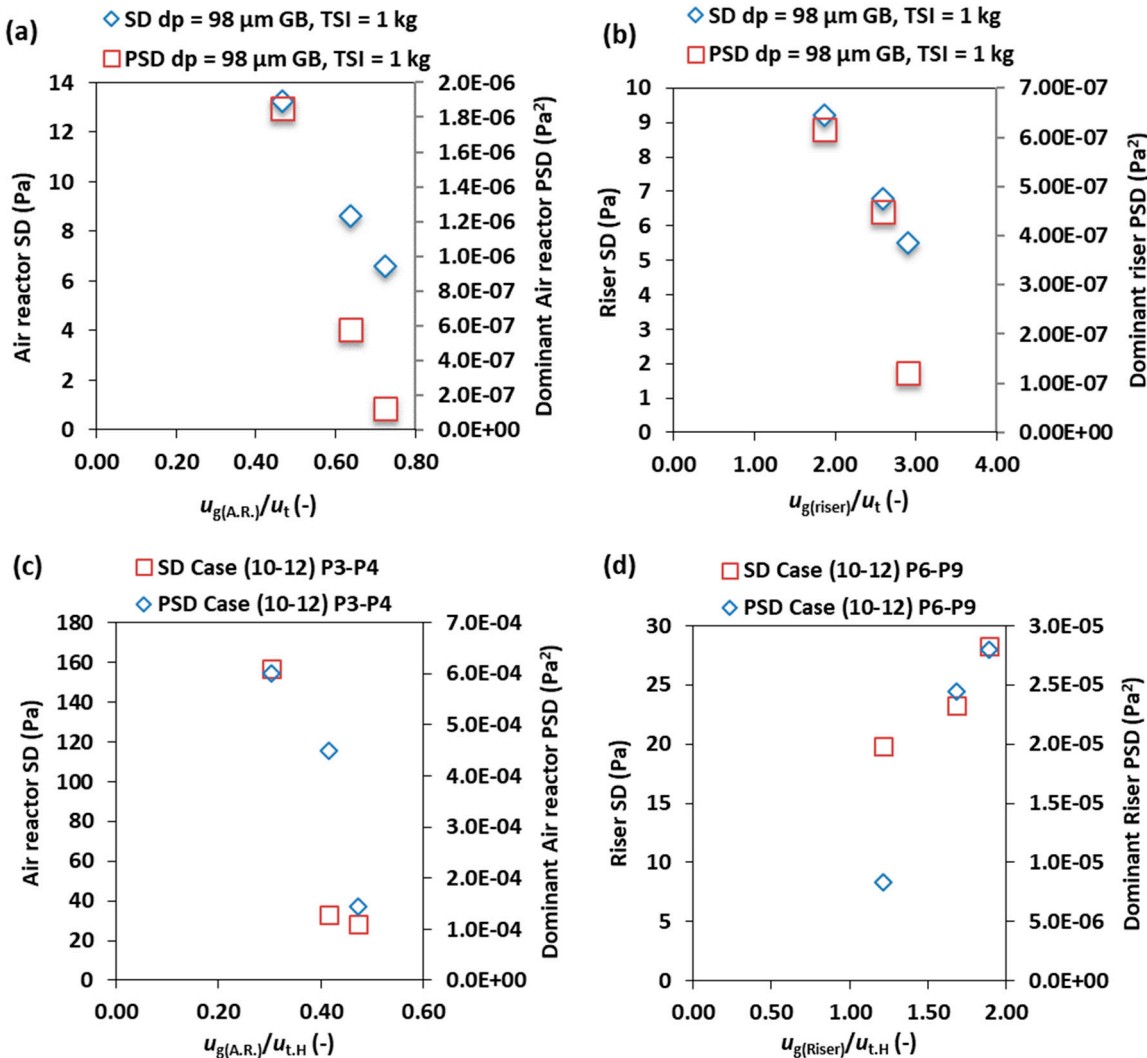

**Figure 7.** Comparison of the *SD* of the pressure fluctuation and the *PSD* analysis on the air reactor and the riser as a function of of $u_{g(A.R.)}/u_{t.H}$ and $u_{g(riser)}/u_{t.H}$ (**a**,**b**) SSS [$d_{GB}$ = 98 µm and TSI = 1 kg] and (**c**,**d**) BMS [10 wt.% of $d_{PE}$ = 231 µm, 90 wt.% of $d_{GB}$ = 116 µm, TSI = 2 kg].

### *3.3. Effect of the Total Solids Inventory*

Increasing total solids inventory affected the air reactor and the riser pressure fluctuation intensity and the dominant *PSD* value for SSS and BMS. For SSS and BMS, at fixed air reactor inlet velocity, the solids holdup and the intensity of the pressure fluctuation were higher at 2 kg. This is the product of increasing the solids inventory at fixed inlet velocity in the air reactor. Even though for example for SSS the solids holdup increased in the air reactor because of increasing the inventory, the solid circulation rate increased from 3.86 g/min, 22.22 g/min to 180.84 g/min for 1 kg, 1.5 kg and 2 kg [52], respectively. Figure 8 shows the effect of increasing the total solids inventory on the *SD* of the pressure fluctuation and the dominant *PSD* on the air reactor and the riser. For both systems, the higher the total inventory of solids, the higher the pressure fluctuation intensity and the dominant *PSD* in the air reactor and the riser.

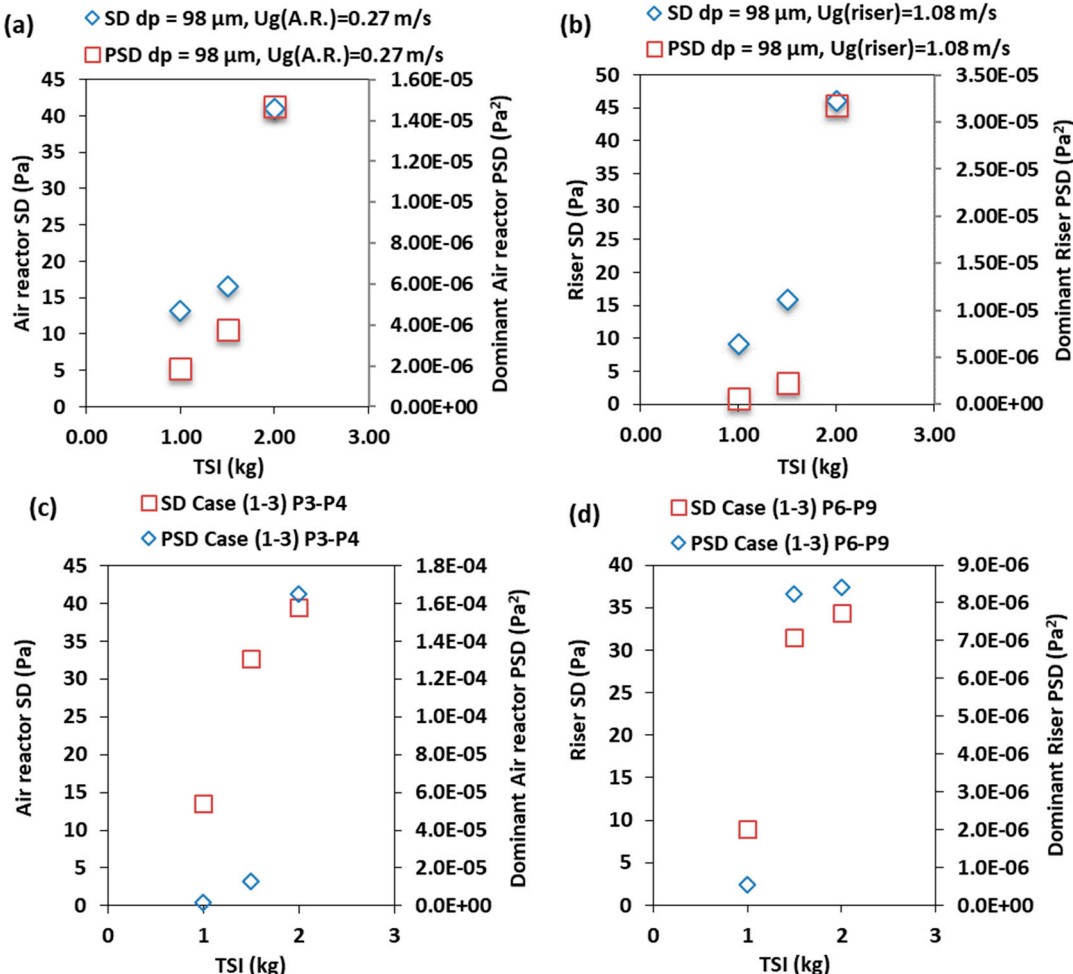

**Figure 8.** Comparison of the *SD* of the pressure fluctuation and the *PSD* analysis in the air reactor and the riser as a function of TSI: (**a**,**b**) SSS [$d_{GB}$ = 98 μm] (**c**,**d**) BMS cases 1–3 [10 wt.% of $d_{PE}$ = 231 μm, 90 wt.% of $d_{GB}$ = 98 μm] at $u_{g(A.R.)}$ = 0.27 m/s and $u_{g(riser)}$ = 1.08 m/s $u_{g(F.R.)}$ = 0.088 m/s.

### 3.4. Effect of the Particle Size

Varying the particle size for SSS influenced the profiles of the pressure and the solids holdup in the air reactor and in the riser [52]. The larger particle (i.e., $d_{GB}$ = 138 μm) has a higher pressure fluctuation intensity and holdup in the air reactor than the smaller particle (i.e., $d_{GB}$ = 98 μm), as reported in Figure 9a. As the particle size increases, $u_{(mf)}$ of the larger particles increases from 0.0085 m/s to 0.012 m/s. As a result, the *PSD* of the dominant amplitude is higher for the larger particles, as shown in Figure 9b. Figure 10, shows the effect of the particle size on the *SD* of pressure fluctuation and the dominant *PSD*, respectively. In the air reactor, the *SD* and *PSD* values are higher for the larger particle at different operating conditions, as shown in Figure 10a,b.

In the riser, the smaller particle has higher pressure fluctuation intensity and its *PSD* amplitude values are also higher, see Figure 9c,d and Figure 10c,d. This can be attributed to the terminal velocity of each particle. The riser velocity to the terminal velocity ratio for the 98 and 138 μm glass beads are 1.88 and 1.08, respectively. Thus, increasing the riser gas to terminal velocities ratio means more solid circulation and solid interaction in the riser. In addition, it can be observed that in the riser, owing to higher $u_{g(riser)}/u_t$ (i.e., by increasing $u_{g(riser)}$ not decreasing $u_t$) the riser's pressure fluctuation *SD* and *PSD* for the large particles become slightly higher (Figure 10c), this is because of the increased presence of the larger particles in the riser.

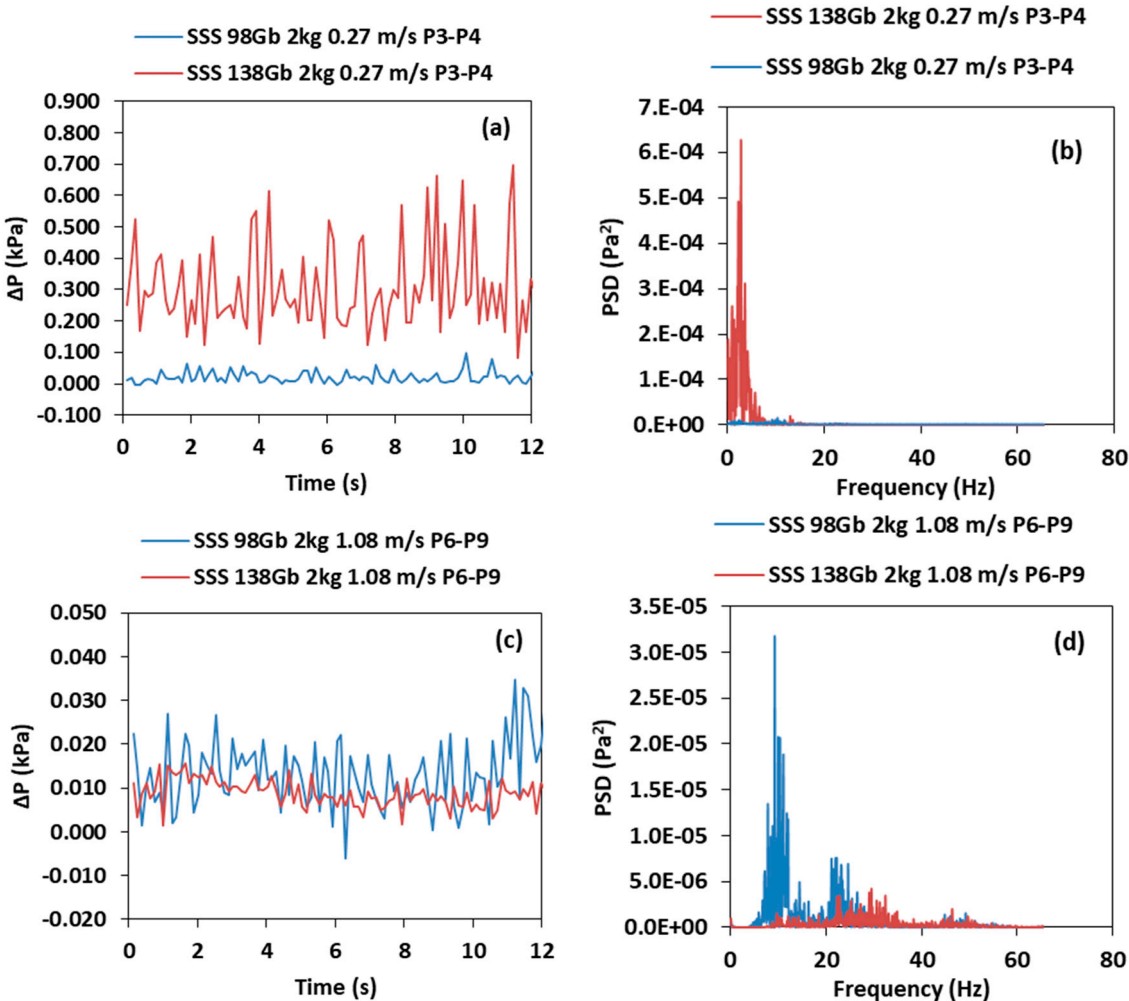

**Figure 9.** Evolution of the pressure fluctuation and the spectral analysis single species systems with 2 kg of $d_{GB}$ = 98 and 138 μm with $u_{g(F.R.)}$ = 0.088 m/s: (**a**,**b**) at $u_{g(A.R.)}$ = 0.27 m/s points 3–4, (**c**) and (**d**) at $u_{g(riser)}$ = 1.08 m/s points 6–9.

By contrast with SSS, for a BMS at the same superficial gas velocity in the air reactor, increasing the particle size by increasing the BMS particle diameter ratio (see Table 3, e.g., cases 2 and 5) leads to a lower pressure drop fluctuation value [23]. This is because of increasing $u_{mf(mixture)}$; thus, as the particle diameter ratio increased from 2.4 to 2.8, less bed expansion and fluidization was observed, as seen in Figure 11a,b. Consequently, the solids holdup of the case with larger diameter ratio near the bed bottom was higher; the dominant *PSD* of the pressure fluctuation were higher with less distribution, which showed as less pressure intensity and *SD* (Figure 12a,b). Conversely, the fluidization of the bed with the smaller diameter ratio is more intense due to a lower $u_{mf(mixture)}$ (i.e., $u_{mf(mixture)}$ = 0.00769 m/s). As a result, intense pressure fluctuation was measured with a wider *PSD* distribution and a higher *SD*, as shown in Figure 11a,b and Figure 12a,b. It seems that from the above observation, the *SD* value was more affected by the fluidization behaviour, while the *PSD* value was more sensitive to and affected by the holdup/presence of the larger diameter particles in the mixture.

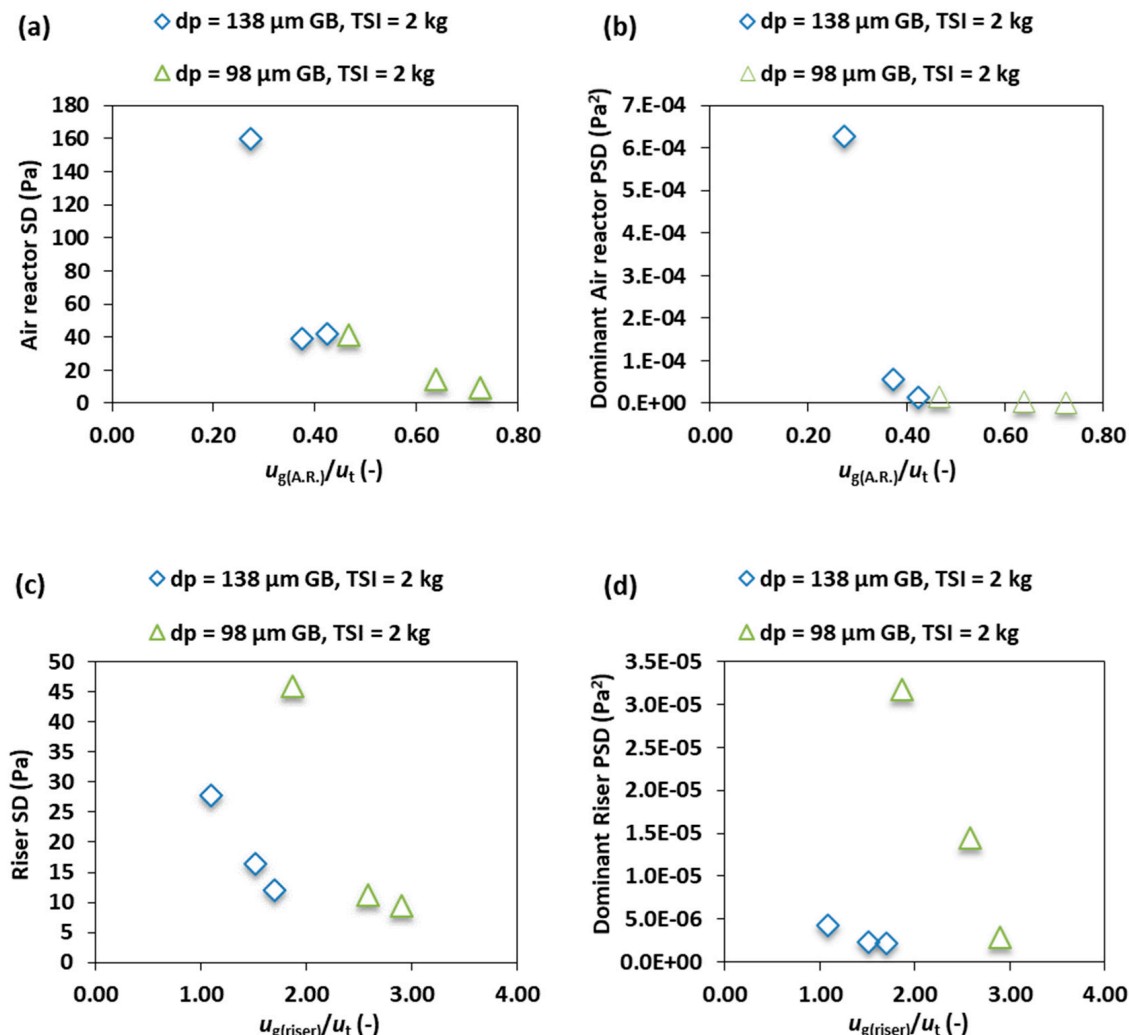

**Figure 10.** Comparison of the *SD* of the pressure fluctuation and the *PSD* analysis on the air reactor (**a**,**b**) and the riser (**c**,**d**) as a function of particle size.

In the riser, the pressure fluctuation value and intensity in terms of *SD* and *PSD* were higher for the mixture with the smaller polyethylene particles, as shown in Figure 11c,d and Figure 12c,d. This is because it has a lower terminal velocity, which leads to a higher solids holdup in the riser (i.e., similar to SSS, Figure 10c,d).

### 3.5. Effect of BMS Composition

Varying the wt.% of each species in the BMS was investigated, and the pressure fluctuation intensity increased as the wt.% of the polyethylene increased in the air reactor. At fixed inlet gas velocity, increasing the wt.% of polyethylene increases the solid circulation rate as well as the percentage of circulated polyethylene more quickly than glass bead particles (polyethylene here is the species with a lower terminal velocity). Consequently, as the polyethylene wt.% increased, the percentage of glass beads in the circulated solids decreased (GB%: 94.4%, 90.4% and 73%) and the holdup of the glass bead particles increased in the air reactor [23]. Smooth frequency and distribution in the air reactor was measured. This can be attributed to the fact that, at this high solid circulation rate, the particles are exiting the air reactor at a higher rate, thus less chaotic interaction and collision of particles occur. Undoubtedly, an increase in the composition of the lower terminal species increases the pressure and the intensity (i.e., *SD*) in the riser (see Figure 13a). The spectral analysis and the distribution in the riser were also smoother and follow the same behaviour as that of the air reactor. This is again due to low

solid interaction in the riser at high solid circulation rate. The dominant amplitude (*PSD*) increases in the riser as the wt.% of the polyethylene in the CFM–CLC increases, as shown in Figure 13b.

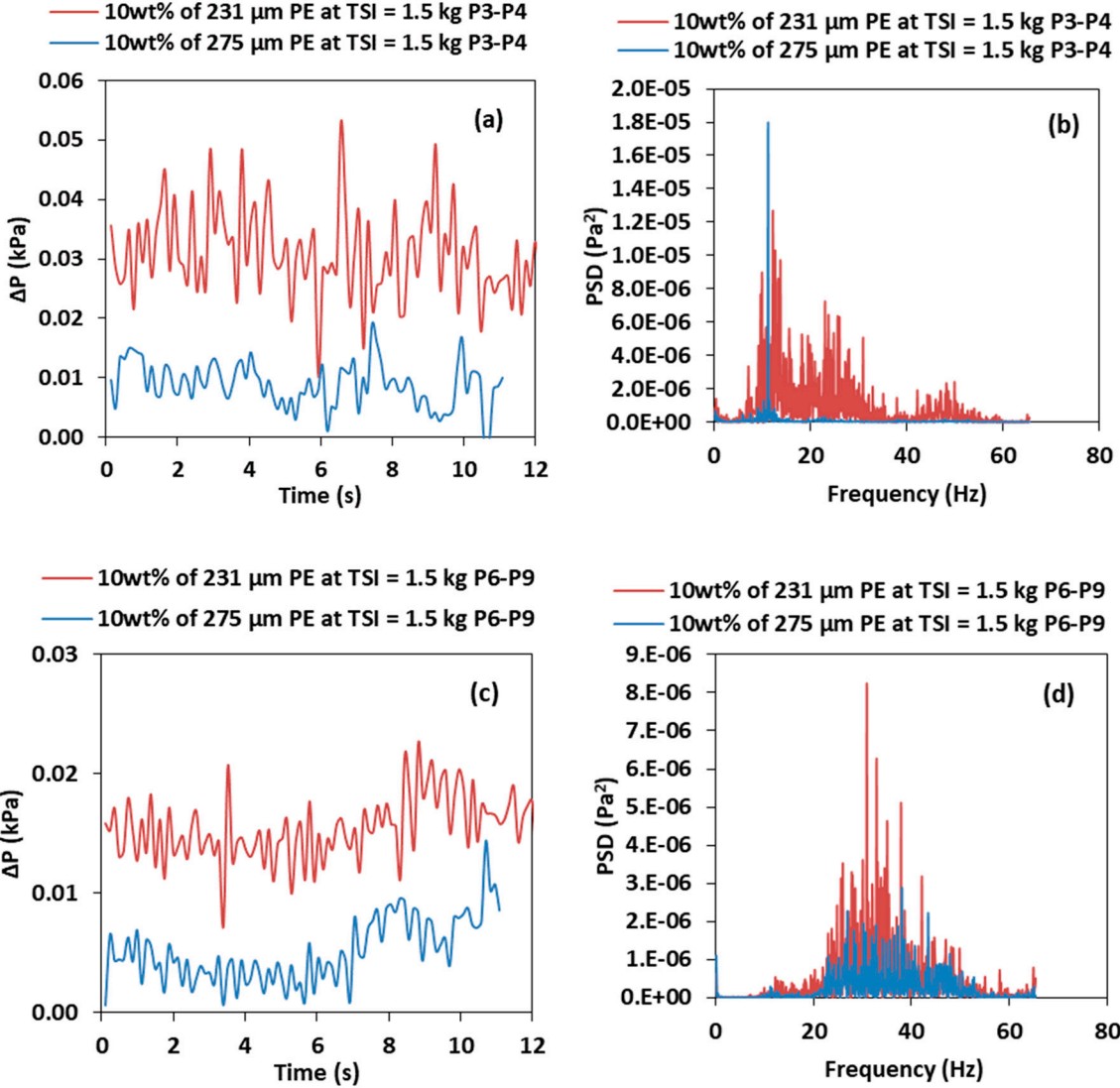

**Figure 11.** Evolution of the pressure fluctuation and the spectral analysis of binary mixture system (BMS) on the air reactor (**a**,**b**) and the riser (**c**,**d**) as a function of particle size [10 wt.% of $d_{PE}$ = 231 and 275 µm, 90 wt.% of $d_{GB}$ = 98 µm, TSI = 1.5 kg, $u_{g(riser)}$ = 1.08 m/s, and $u_{g(F.R.)}$ = 0.088 m/s].

This information of the pressure fluctuation and the *PSD* analysis are very useful if being used along with the *SD* analysis to understand the consequence of changing one variable in the initial operating condition of CFBs when SSS and BMS are utilized.

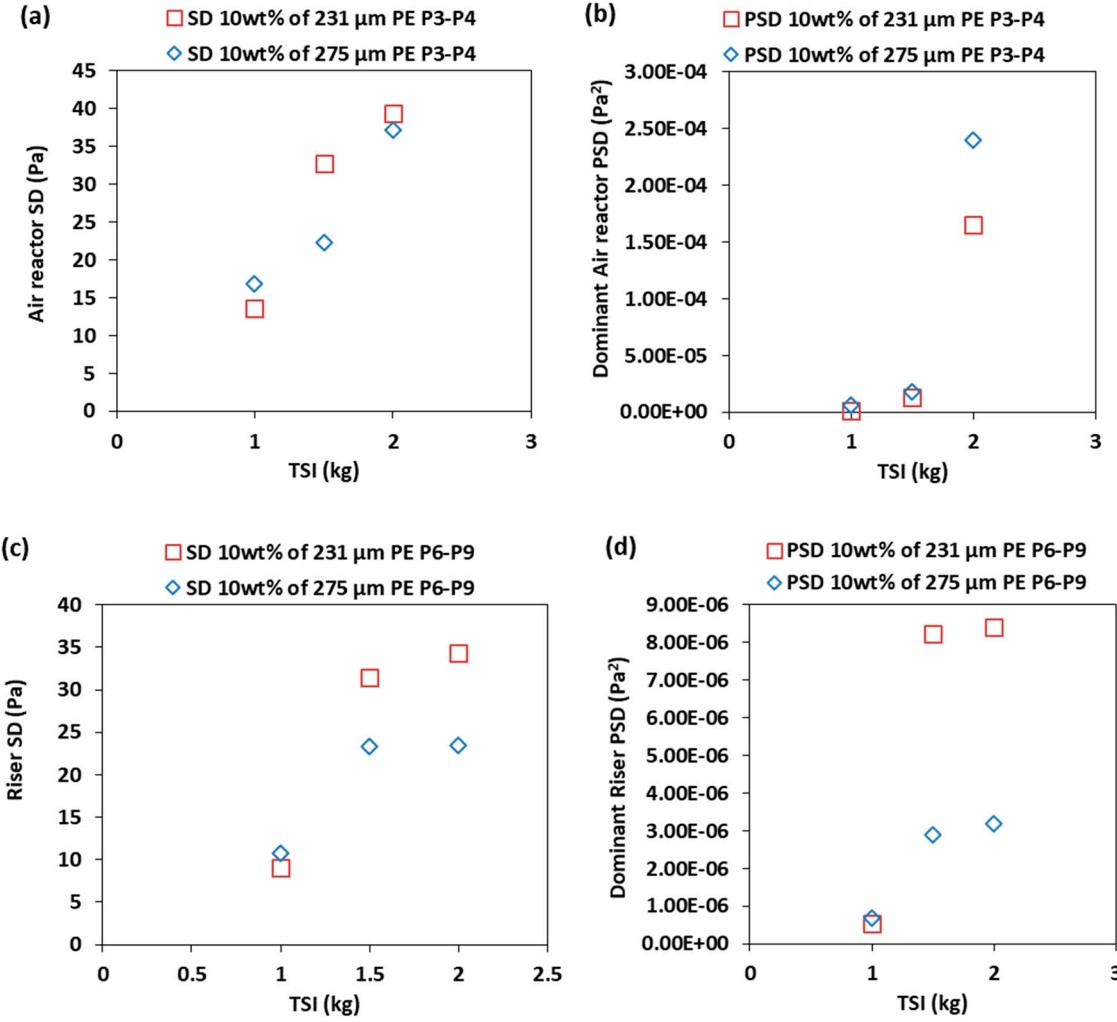

**Figure 12.** Comparison of the *SD* of the pressure fluctuation and the *PSD* analysis on the air reactor (**a**,**b**) and the riser (**c**,**d**) as a function of particle size [10 wt.% of $d_{PE}$ = 231 and 275 μm, 90 wt.% of $d_{GB}$ = 98 μm, TSI = 1, 1.5 and 2 kg, $u_{g(\text{riser})}$ = 1.08 m/s, and $u_{g(\text{F.R.})}$ = 0.088 m/s].

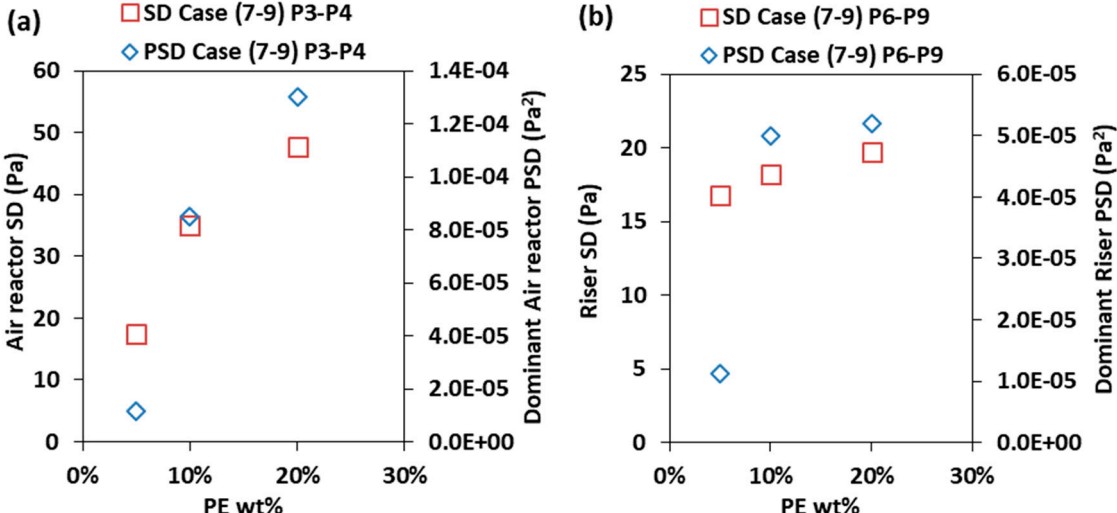

**Figure 13.** Comparison of the *SD* of the pressure fluctuation and the *PSD* analysis (**a**) on the air reactor and (**b**) the riser as a function of composition [$d_{PE}$ = 231 μm, $d_{GB}$ = 138 μm, $u_{g(\text{riser})}$ = 1.68 m/s and $u_{g(\text{F.R.})}$ = 0.088 m/s].

The observed behaviour for the above tested conditions are summarized in Table 4. It can be seen that there is clear difference between the two systems (i.e., SSS and BMS). In general, in BMS the *SD* values were found to be influenced by the fluidization regime and initial operating condition, the *PSD* values were more sensitive and affected by the presence of the particles with the higher terminal velocity in the mixture. Further investigations are needed to fully understand the riser pressure fluctuation when BMS is used, specially the hydrodynamic of such a chaotic region in CFB. Similar studies could prove beneficial to identifying the mixing region, and the critical region at which the particles gained enough momentum to circulate out of the riser [53].

**Table 4.** Summary of the effect of changing the operating variables on SSS and BMS.

| Effect of Increasing | SSS | | | | BMS | | | |
|---|---|---|---|---|---|---|---|---|
| | Air Reactor | | Riser | | Air Reactor | | Riser | |
| | SD | PSD | SD | PSD | SD | PSD | SD | PSD |
| $u_{g(A.R.)}$ and $u_{g(riser)}$ | ↓ | ↓ | ↓ | ↓ | ↓ | ↓ | ↑ | ↑ |
| TSI | ↑ | ↑ | ↑ | ↑ | ↑ | ↑ | ↑ | ↑ |
| $d_p$ | ↑ | ↑ | ↓ | ↓ | ↓ | ↑ | ↓ | ↓ |
| BMS Wt.% (5, 10, 20%) | - | - | - | - | ↑ | ↑ | ↑ | ↑ |

### 3.6. Effect of the Fuel Reactor Fluidization Velocity

The effect of changing the fuel reactor's superficial fluidization gas velocity on the overall pressure profile, pressure fluctuation and the solid circulation rate were investigated. Figure 14 shows the pressure profile of the BMS, which consisted of 90 wt.% of $d_{GB}$ = 138 μm and 10 wt.% of $d_{PE}$ = 328 μm at constant $u_{g(riser)}$ = 1.08 m/s, while altering the fluidization velocity in the fuel reactor in a range of $u_{g(F.R.)}$ = 0.0294–0.147 m/s. It can be observed that the overall pressure profiles were the same for different fuel reactor fluidization velocities, especially in the air reactor and the riser (i.e., points 2–5 and 6–9 in Figure 1). Still, there was a slight decrease in the fuel reactor pressure (i.e., point 17) as the fluidization velocity was increased owing to the decrease in the bed's material height.

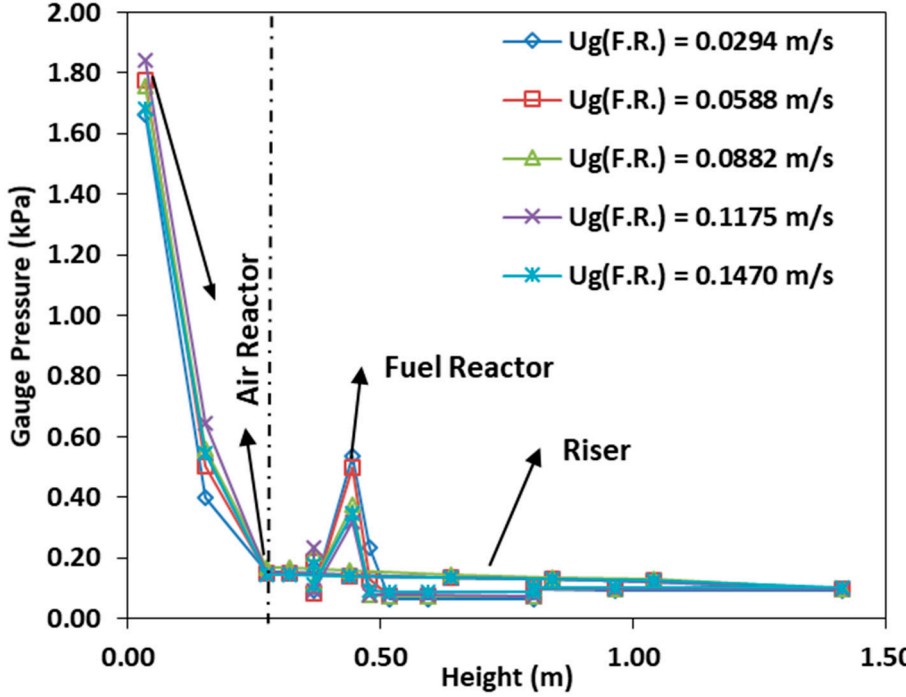

**Figure 14.** CFM–CLC pressure profile for BMS [10 wt.% $d_{PE}$ = 328 μm, 90 wt.% $d_{GB}$ = 138 μm, $u_{g(riser)}$ = 1.08 m/s and $u_{g(F.R.)}$ = 0.0294–0.147 m/s].

However, varying the fuel reactor fluidization velocity had no effect on the riser pressure fluctuation, as shown in Figure 15a. Additionally, it did not influence the overall solid that circulated in the system (see Figure 15b). At each fuel reactor fluidization velocity, the solid circulation rate measurement was repeated five times. The averaged solid circulation rate of each fuel reactor fluidization velocity is shown in Figure 15b as a symbol, and the overall average for all the measured values is shown as a dotted line. It can be seen that the solid circulation rate was almost around the same value; the majority of the experiments were around 24–25 g/min with ±5% deviation. Therefore, the results indicate that the air reactor and the riser but not the fuel reactor were the main CLC components influencing the global circulation of solids throughout the CLC system.

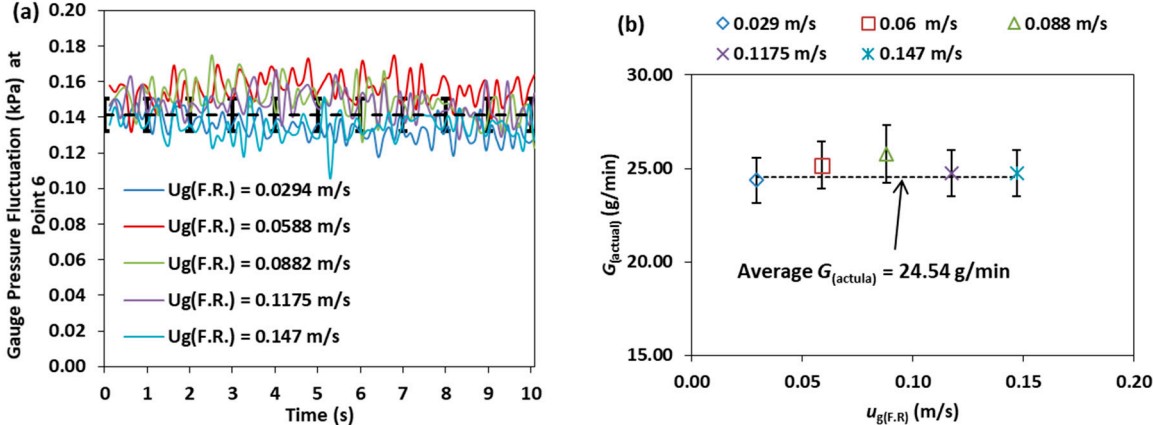

**Figure 15.** (**a**) Riser pressure fluctuation at point 6 against the fluidization velocity in the fuel reactor, and (**b**) the solid circulation rate in the CLC system as a function of $u_{g(F.R.)}$ at constant $u_{g(A.R.)}$.

## 4. Conclusions

The current work reports on the systematic study of the pressure fluctuation in the riser of a 10 kW$_{th}$ CFM–CLC was conducted using SSS and BMS. Quantitative and qualitative analysis were provided to understand the effect of changing variables such as air/fuel reactor superficial gas velocities, total solids inventory, the particle size and BMS initial composition on the pressure measurements and its fluctuation. It was observed that the air reactor and the riser are the main components driving the global solid circulation rate. Typically, the dominant amplitude of the pressure fluctuation is commonly used as a tool in investigating the hydrodynamics of CFB (i.e., fluidization regimes characteristics). However, even though the information obtained from the amplitude of the pressure fluctuation is valuable, it cannot be useful unless studied systematically by observing the effect of varying the initial operating conditions. Therefore, further analysis was provided in this work to understand the effect of changing important variables on the riser hydrodynamic in term of time series pressure fluctuation analysis, spectral analysis, and comparison between the dominant *PSD* and *SD* of the pressure fluctuation.

With the focus on the current studied conditions, the following points were observed:

- The introduction of a second species into the system influenced the intensity of the pressure fluctuation in the air reactor and the riser. In general, the pressure fluctuation in terms of *SD* and *PSD* are higher for BMS when compared to SSS. Increasing the wt.% (i.e., the composition of PE) increases the intensity of the pressure fluctuation and the percentage of the second species in the solid circulation rate.

- For SSS, the solids holdup and interaction of particles increases as the velocity decreases in the air reactor, which lead to a higher *SD* of the pressure fluctuation and *PSD* amplitude. Increasing the riser velocity increases the solids holdup on the riser; however, the *SD* and *PSD* amplitude decreases because more solids are transported upwards at a higher velocity and fewer solids fall downwards. Similarly, as the air reactor superficial gas velocity decreases in the BMS, the solids

holdup increases of both species, therefore, higher *SD* of the pressure fluctuation and dominant *PSD* are observed. However, in the BMS, increasing the riser velocity increases the solids holdup of the larger particles in the riser. Thus, this will lead to increasing the *SD* and *PSD* of the dominant amplitude in the riser, which will follow the opposite trend in comparison with SSS.

- Increasing the total solids inventory for the SSS and BMS increases the pressure fluctuation in the air reactor and the riser of the CLC–CFM, because of increasing the solids holdup at constant superficial gas velocity.
- As the diameter of the particle increases the pressure fluctuation intensity, the frequency distribution and the *PSD* amplitude in the air reactor increases. Conversely, it decreases in the riser because the smaller particles are transported at a higher rate into the riser than the larger particles because of the terminal velocity differences.
- Varying the fuel reactor fluidization velocity at constant air reactor fluidization velocity does not influence the air reactor and riser pressure profile, the intensity of the pressure fluctuation and the solid circulation rate.

**Author Contributions:** Conceptualization, Y.A.A., B.M. and E.D.; Formal analysis, Y.A.A. and C.L.; Funding acquisition, Y.A.A., B.M. and E.D.; Investigation, Y.A.A.; Methodology, Y.A.A.; Resources, B.M.; Supervision, Z.P., B.M. and E.D.; Validation, Y.A.A.; Visualization, Y.A.A.; Writing—original draft, Y.A.A., Z.P. and Z.A.; Writing—review and editing, Y.A.A., Z.P., Z.A., B.M. and E.D.

**Funding:** This research and the preparation of the manuscript was funded and supported by the ministry of education (MoE) of the Kingdome of Saudi Arabia, by Moits Pty Ltd., New South Wales Coal Innovation, and the Australian Research Council (LP100200872 and MOC10/1067), Sustainable Energy Technologies centre at KSU for open access fund, respectively.

**Acknowledgments:** The author extends his appreciation to Deputy of Research and Innovation DRI-MOE at the Kingdom of Saudi Arabia for the Postdoctoral Fellowship Program (PFP), also acknowledge the Deanship of Scientific Research (DSR), King Saud University for their support during this program. Also, the author wishes to acknowledge the facility support by the University of Newcastle (Australia), The authors would as well like to thank Ron Robert and Mr Neil Gardner for their support in setting up the experimental facility and resolving related technical issues.

**Conflicts of Interest:** The authors declare no conflict of interest. The funders had no role in the design of the study; in the collection, analyses, or interpretation of data; in the writing of the manuscript, or in the decision to publish the results.

## Nomenclature

### Symbols

| | |
|---|---|
| $Ar$ | Archimedes number |
| $d_{\text{p}}$ | particle mean diameter (m) |
| $d_{\text{in(riser)}}$ | riser inner diameter (mm) |
| $d_{\text{in(A.R.)}}$ | air reactor inner diameter (mm) |
| $g$ | gravitational acceleration (m/s$^2$) |
| $G_{\text{actual}}$ | actual solid circulation rate (g/min) |
| $h$ | height (mm) |
| $N$ | number of the sampling points |
| P | monitored period (s) |
| $Re_{\text{dp(A.R.)}}$ | particles Reynolds number in the air reactor (–) |
| $Re_{\text{dp(riser)}}$ | particles Reynolds number in the riser (–) |
| $u_{\text{mf}}$ | the minimum fluidization velocity (m/s) |
| $u_{\text{mf(mixture)}}$ | the minimum fluidization velocity of binary mixture (m/s) |
| $u_{\text{t}}$ | particle terminal velocity (m/s) |
| $u_{\text{t.H}}$ | terminal velocity of the particle with the higher $u_{\text{t}}$ in the binary mixture (m/s) |

| $u_{t.L}$ | terminal velocity of the particle with the lower $u_t$ in the binary mixture (m/s) |
| $u_g$ | interstitial riser gas velocity (m/s) |
| $u_{g(riser)}$ | riser gas velocity (m/s) |
| $u_{g(A.R.)}$ | air reactor gas velocity (m/s) |
| $u_{g(F.R.)}$ | fuel reactor gas velocity (m/s) |
| $u_{mf(mixture)}$ | binary mixture minimum fluidization velocity (m/s) |
| $t$ | time (s) |

**Greek letters**

| $X$ and $Y$ | amplitude at the angular frequency |
| $\Delta h$ | height between the two pressure ports (m) |
| $\Delta P$ | pressure drop between two points (kPa) |
| $\Delta P_{Average}$ | average pressure drop between over the entire sample (kPa) |
| $\rho_f$ | fluid density (kg/m$^3$) |
| $\rho_p$ | solid or particle density (kg/m$^3$) |
| $\rho$ | density (kg/m$^3$) |
| $\phi/\phi s$ | solids holdup /solids holdup (−) |
| $\mu$ | fluid viscosity (Pa. s) |
| $\omega$ | angular frequency (rad/sec) |

**Abbreviations**

| BMS | Binary-mixture system |
| CFB | Circulating fluidized bed |
| CFM | Cold-flow model |
| CLC | Chemical looping combustion |
| GB | Glass bead |
| PE | Polyethylene |
| *PSD* | Power spectrum density (Pa$^2$) |
| SSS | Single-species system |
| *SD* | Standard deviation (Pa) |
| TSI | Total solids inventory (kg) |

**Appendix A**

*Steady State Operation*

The results reported in this work were obtained under steady state operating conditions. The running time required to reach steady state was determined by observing the variation in the weight percentage of polyethylene species in the binary mixture of polyethylene and glass beads within the air reactor, fuel reactor and the loop seals as a function of time as shown in Figure A1. An explanation and more details were provided in earlier work [23].

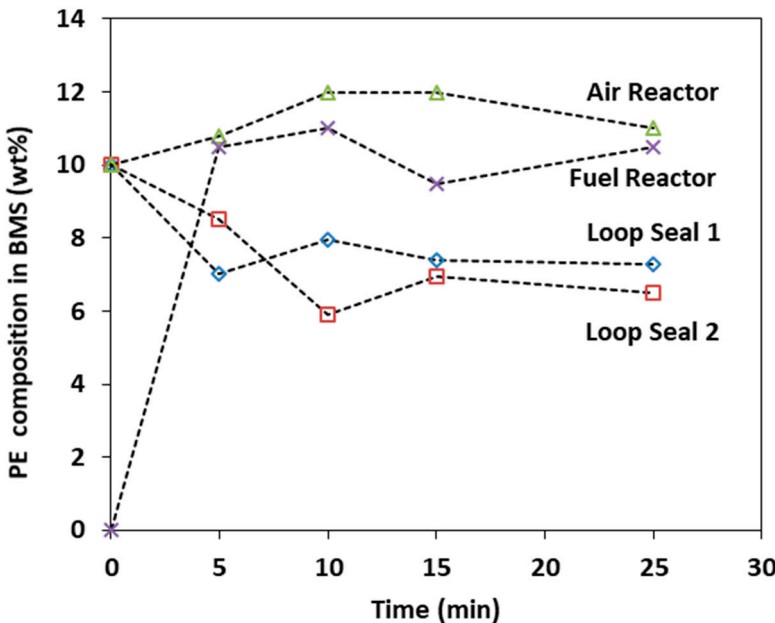

**Figure A1.** Polyethylene (PE) composition evolution in BMS as a function of time, 10 wt.% polyethylene [TSI = 1 kg, $d_{PE}$ = 231 μm, $d_{GB}$ = 116 μm, $u_{g(riser)}$ = 1.6 m/s, $u_{g(F.R.)}$ = 0.088 m/s ] Reproduced with permission from [Alghamdi et al.], [Chemical Engineering journal]; published by [Elsevier], [2013] [23].

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
