# Peer review of "Systematic Study of Pressure Fluctuation in the Riser of a Dual Inter-Connected Circulating Fluidized Bed: Using Single and Binary Particle Species"

_processes, doi:10.3390/pr7120890_

Round 1

Reviewer 1 Report

Systematic Study of Pressure Fluctuation in the Riser of a Dual Inter-Connected Circulating Fluidized Bed: Using Single and Binary Particle Species

Yusif A. Alghamdi, Zhengbiao Peng, Caimao Luo, Zeyad Almutairi, Behdad Moghtaderi, Elham Doroodchi

Review

Though I think you have something good here, it is sadly lost to an information overload. You need to boil down your results to the bare essence of what you want to communicate with the paper. There is way too much of both text and figures in the results chapter and the structure could be better. I suggest the following to improve this:

Present the reader with a clear aim of the investigation. I understand that it is interesting to you what pressure fluctuations are in the riser of this unit, but you need to formulate it with the reader in mind. What is interesting about this paper to someone else? Every line and figure need to contribute toward this aim, or it can be removed from the paper. Think about what is new here and focus on that. For example, it is well known that increasing the gas velocity reduces pressure fluctuations. This is not something you need to show the reader in any other way than telling them you have verified this for your experiments and show a short example. There are more things like this throughout the results. Instead, if some unexpected trends are seen it is worth spending time on showing those. What does the frequency analysis show here that the standard deviation does not? If they show the same thing (i.e. decreasing amplitude at certain frequencies indicate smaller fluctuations, but so does a smaller standard deviation). It seems one of these measures could be all but taken out of the paper. Focus on the comparison between SSS and BMS. As it is now, you present them separately but I think much of the value of this paper lies in the comparison. I suggest that you reorganize the paper to show a comparison between these two for each variable instead of having separate chapters for them.

General:

It gets a little confusing and hard to read the text with such frequent use of acronyms. Please consider writing the full sequence of words much more often. u_t,L missing from nomenclature The figures are very hard to read due to the appearance (shadows etc.) and thickness of the lines. Please use another software to make them, or at least go with a more sober theme and thinner lines. Perhaps also make some lines dashed to aid those with color blindness. In figures with height on the x-axis, please include a vertical line marking the transition from AR to riser. You talk a lot about increasing and decreasing trends in the results section. It becomes very difficult to keep straight what influences what and how. I suggest you make a table where you show with arrows what the different trends are for different parameters.

L44-45: I’d say that the application of CLC lies between these two points, i.e. it is both part of the chemical engineering application and in energy conversion/steam production.

L80-83: Please give some references at each point

L100: “altered from 1 to 2…” → “varied between 1 to 2…”

L120ff: A more thorough presentation of the scaling parameters needs to be made.

What are the scaling ratios here? How is the gas/solids density scaling ratio maintained for both CuO and SiO2, which have such different densities? Do you use gases different than air in the system, or is only parts of the temperature range of 500-1000°C only valid for specific bed materials? Which set of scaling laws was applied?

L136: Why was only such a small amount of data points used for averaging? It would represent only 0.2 s with your 500 Hz sampling frequency.

L150: Did you also drain some fine particles from the system to maintain the same TSI or did the TSI increase from the SSS to BMS cases?

L165: Does ф here represent the solids hold-up? You talk about pressure in the text around the equation, which makes it confusing.

L170: You previously stated 10-60 s and here 60 s, which one is correct?

L218: Instead of the figures showing the pressure as a function of times, I suggest that you show SD as a function of ug, and include all different TSI in one figure. Likewise you could show the dominant frequency as a function of ug for the different cases. Though it can be good to keep just one example of a p(t) and full PSD. The paper is suffering from a great abundance of figures, which makes it hard to keep track of the different cases. That you put some figures in appendix does not help since you still analyze them in the text (in fact it just makes the reader go back and forth between text and appendix, which is counterproductive).

L280: Figure 6c, there seems to be some disturbance of the pressure at 30 s. Could it be something external to the system affecting the pressure? I would advise you to remove data from 30-60 s from both 6c and 6d.

320: The sample time is very low. Why not use 60 s as for SSS? Data is present near 0 Hz, which indicates that some low frequencies maybe not dissolved properly. This continues in all PSD figures in this chapter. See e.g. Fig. 12d where almost no data can be read from the figure.

375: Figure names (e.g. a, b, c) are in a different position from previous figures.

Reviewer 2 Report

In this manuscript, the authors studied the pressure fluctuation in the air reactor and riser of a CFM-CLC system when two different kinds of particles are circulated in the system. The authors found that the existence of a second kind of particle increases the fluctuation of system. The reviewer recommend major modification before the manuscript can be published.

The authors should better justify the motivation of the study. In line 48 of the manuscript, ref 3 and 4 use a binary metal oxide system, but not necessary two particle that differ in size and/or density. In fact, the binary metal oxide used in CLC are usually used in the form of a single particle containing both ingredients. The use of two kinds of particles that differ in size and/or density in a CLC system is not common.

At line 116: Dimensionless analysis should be further illustrated. What dimensionless numbers are compared? The authors claimed that GB and PE particles at room temperature can represent a very wide range of different oxygen carrier materials over a wide range of temperature. This will need to be justified.

In terms of signal processing, at line 170: According to the authors, the sampling frequency is 500 Hz, which should correspond to a PDS up to that frequency. However, in all figures for spectral analysis, the frequency only goes up to 65 Hz. What happened to the higher frequency components?

As a research in fluid dynamics, the analysis in the manuscript should be presented in terms of dimensionless numbers/groups. This will allow other researches to apply the findings in systems at different scales. The authors should consider reorganizing the data in terms of dimensionless numbers/groups.

Other minor comments:

The quality of the figures should be improved. The lines and symbols in the current figures overlays each other. Line 58: ϕs need to be defined. Abbreviations need to be defined the first time they show up. PSD, TSI, SCR, etc. Line 132: How is SCR measured? How is electric static handled in the CFM? Plastic and glass particles are prone to be affected by electric static. Line 167: The following sentence is not clear: “in which t denotes a monitored variable (representing monitored against the temporally varying pressure),” Please modify or further illustrate. “t” usually denotes time, though the authors state otherwise. Please include “t” in the Nomenclature section if it’s not “time”. Figure 3, 5, etc. Is it possible to provide the error bar for SD analysis? Comparing Fig 4 and Fig 5 is confusing. For example, compare Fig 4(c) and Fig 5(b), with particle 98GB and gas velocity of 1.68m/s in the riser. Fig 4(c) shows that the pressure fluctuation at TSI of 2kg should be much larger than that of the smaller TSIs. However, in Fig 5(b), the SD at TSI=2kg is smaller than that at TSI=1.5kg. Why? Fig 9(a) and 9(c): Data for TSI=1.5kg is missing. Line 498: “Increasing the riser velocity increases the solids holdup on the riser and the pressure fluctuation.” Based on the SD analysis, increasing the riser velocity decreases the SD. Why does the authors states that the pressure fluctuation is increased?

Round 2

Reviewer 1 Report

Review round 2

L134: The abrasion effect should be of little significance unless you are using optical measurement methods from the outside of the reactor. There are plenty of cold flow models that use sharp sand as bed material without issues. Furthermore, many of the oxygen carriers you list are softer than glass beads (although more sharp).

L139: Please make a reference to your other paper, and that’s where the scaling procedure is explained.

L139: This scaling procedure is rather unorthodox, and as you mention in the other paper it needs more verification in units with different sizes, design parameters before it can be considered generally applicable. Please clarify that this is not a generally accepted scaling procedure, and that it only applies specifically to your equipment, but also clarify that it is validated for this unit. Otherwise other readers might become confused as to the validity of your results. Also please give the scale of the unit.

L141: The equation looks to have been corrupted and is missing some characters

Thank you for taking my previous comments into consideration and restructuring the paper a bit. I think it became much more clear after that.

Reviewer 2 Report

Recommend for publication
